# CONTROLLABLE VIDEO GENERATION WITH PROVABLE DISENTANGLEMENT

**Yifan Shen**[1]***Peiyuan Zhu**[1]***Zijian Li**[1,2]**, Shaoan Xie**[2]**, Namrata Deka**[2]**,**
**Zongfang Liu**[1]**, Zeyu Tang**[2]**, Guangyi Chen**[1,2]**, Kun Zhang**[1,2]†

[1]Mohamed bin Zayed University of Artificial Intelligence
[2]Carnegie Mellon University

## ABSTRACT

Controllable video generation remains a significant challenge, despite recent advances in generating high-quality and consistent videos. Most existing methods for controlling video generation treat the video as a whole, neglecting intricate fine-grained spatiotemporal relationships, which limits both control precision and efficiency. In this paper, we propose **Co**ntrollable **V**ide**o G**enerative **A**dversarial **N**etworks (CoVoGAN) to disentangle the video concepts, thus facilitating efficient and independent control over individual concepts. Specifically, following the **minimal change principle**, we first disentangle static and dynamic latent variables. We then leverage the **sufficient change property** to achieve component-wise identifiability of dynamic latent variables, enabling disentangled control of video generation. To establish the theoretical foundation, we provide a rigorous analysis demonstrating the identifiability of our approach. Building on these theoretical insights, we design a **Temporal Transition Module** to disentangle latent dynamics. To enforce the minimal change principle and sufficient change property, we minimize the dimensionality of latent dynamic variables and impose temporal conditional independence. To validate our approach, we integrate this module as a plug-in for GANs. Extensive qualitative and quantitative experiments on various video generation benchmarks demonstrate that our method significantly improves generation quality and controllability across diverse real-world scenarios.

## 1 INTRODUCTION

Video generation (Vondrick et al., 2016; Tulyakov et al., 2018; Wang et al., 2022) has become a prominent research focus, driven by its wide-ranging applications in fields such as world simulators (OpenAI, 2024), autonomous driving (Wen et al., 2024; Wang et al., 2023), and medical imaging (Li et al., 2024a; Cao et al., 2024). In particular, controllable video generation (Zhang et al., 2025) is essential for advancing more reliable and efficient video generation models. Despite the impressive results achieved by recent commercial or large-scale models such as Kling (Kuaishou Technology, 2024) and Wan (Wan et al., 2024), precise control over specific aspects of generated video remains a significant challenge, as illustrated in Figure 1. This issue may arise because these models typically represent the video as a unified 4D spatiotemporal block and apply conditioning signals (e.g., text prompts) directly to this global representation. Recent approaches (Ho et al., 2022; Zhou et al., 2022; Yang et al., 2024b; Zheng et al., 2024) follow a similar strategy, with differences in modeling frameworks (e.g., diffusion or VAE) and shapes (e.g., vectors or spatiotemporal blocks). However, these formulations often neglect the intricate spatio-temporal structure of videos, thereby limiting the ability to disentangle and control fine-grained factors, such as head movements and eye blinking.

To address this issue, one intuitive solution is to learn a disentangled representation of the video, within which the internal relationships are often not considered. Some models (Hyvärinen & Oja, 2000; Tulyakov et al., 2018; Yu et al., 2022; Skorokhodov et al., 2022; Wei et al., 2024) explicitly decompose video generation into two parts: motion and identity, representing dynamic and static

---

*Equal contribution.
†Correspondence to: <kunz1@cmu.edu>.

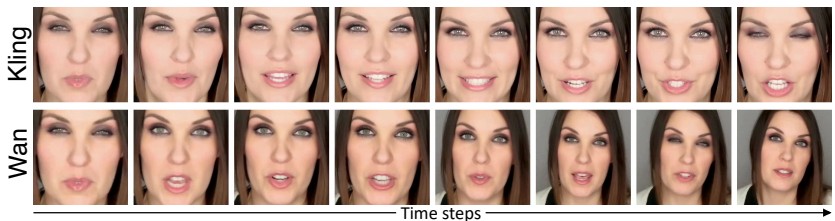

Figure 1: Videos are generated using Kling and Wan with the prompt: "while this person was speaking, the head gradually shifted from the middle to the right." The first row shows that essential motion cues are partially omitted, while in the second row the head size changes undesirably.

information, respectively. This separation allows for more targeted control over each aspect, making it possible to modify the motion independently without affecting the identity. (Zhang et al., 2025; Shen et al., 2023) leverage attention mechanisms to further disentangle different concepts within the video, enhancing the ability to control specific features with greater precision. (Fei et al., 2024; Lin et al., 2023) utilize Large Language Models to find the intricate temporal dynamics within the video and then enrich the scene with reasonable details, enabling a more transparent generative process. These methods are intuitive and effective, yet they lack a solid guarantee of disentanglement, making the control less predictable and potentially leading to unintentional coupling of different aspects.

These limitations of previous approaches motivate us to rethink the paradigm of video generation. Inspired by recent advancements in nonlinear Independent Component Analysis (ICA) (Hyvarinen & Morioka, 2017; Khemakhem et al., 2020; Yao et al., 2022; Hyvarinen & Morioka, 2016) and the successful applications like video understanding (Chen et al., 2024), we propose the Controllable Video Generative Adversarial Network (CoVoGAN) with a Temporal Transition Module plugin. Building upon StyleGAN2-ADA (Karras et al., 2020), we distinguish between two types of factors: the dynamic factors that evolve over time, referred to as **style dynamics**, and the static factors that remain unchanged, which we call **content elements**. We also distinguish between different components within the dynamics, allowing for more precise control. By leveraging the minimal change principle, we demonstrate their block-wise identifiability (Von Kügelgen et al., 2021; Li et al., 2024b) and find the conditions under which motion and identity can be disentangled, explaining the effectiveness of the previous line of video generation methods that separate motion and identity. In addition, we employ sufficient change property to disentangle different concepts of motion, such as head movement or eye blinking. Specifically, we introduce a flow (Rezende & Mohamed, 2015) mechanism to ensure that the estimated style dynamics are mutually independent conditioned on the historical information. Furthermore, we prove the component-wise identifiability of the style dynamics and provide a disentanglement guarantee for the motion in the video.

We conduct both quantitative and qualitative experiments on various video generation benchmarks. For quantitative evaluation, we use FVD (Fréchet Video Distance) (Unterthiner et al., 2019) to assess the quality of the generated videos. We also compare SAP (Kumar et al., 2017), modularity (Ridgeway & Mozer, 2018) and MCC to demonstrate the disentanglement capability. For qualitative analysis, we evaluate the degree of disentanglement by manipulating different dimensions of the latent variables and comparing the resulting video outputs. Experimental results demonstrate that our method significantly outperforms other video generation models of a scale similar to CoVoGAN.

**Key insights and contributions** of our research include:

- We propose a Temporal Transition Module to achieve a disentangled representation, which leverages the minimal change principle and sufficient change property.
- We implement the Module in a GAN, i.e., CoVoGAN, to learn the underlying generating process from video data with disentanglement guarantees, enabling more precise and interpretable control.
- To the best of our knowledge, this is the first work to provide an identifiability theorem in the context of video generation. This helps to clarify previous intuitive yet unproven techniques and suggests potential directions for future exploration.
- Extensive evaluations across multiple datasets demonstrate the effectiveness of CoVoGAN, achieving superior results in terms of both generative quality and controllability.

The remainder of this paper is organized as follows. Section 2 formalizes the video generation process and its identifiability, and discusses how this enables disentangled control over video generation. Sec-

tion 3 provides theoretical insights into when and how block-wise and component-wise identifiability can be achieved. Section 4 introduces the proposed CoVoGAN model and explains how its design is grounded in the theoretical results. Section 5 presents extensive experiments to comprehensively evaluate CoVoGAN. Finally, Section 6 concludes the paper.

## 2 PROBLEM SETUP

### 2.1 GENERATING PROCESS

Consider a video sequence $V = \{\mathbf{x}_1, \mathbf{x}_2, \ldots, \mathbf{x}_T\}$ consisting of $T$ consecutive frames. Each frame $\mathbf{x}_t \in \mathbb{R}^{n_x}$ is generated via an arbitrary nonlinear mixing function $g$, which maps a set of latent variables to the observed frame $\mathbf{x}_t$. The latent variables are decomposed into two distinct parts: $\mathbf{z}_t^s \in \mathbb{R}^{n_s}$, capturing the **s**tyle dynamics that evolve over time, and $\mathbf{z}^c \in \mathbb{R}^{n_c}$, encoding the **c**ontent variables that remain consistent across all frames of the video. Furthermore, these latent variables are assumed to arise from a stationary, non-parametric, time-delayed causal process.

As shown in Figure 2 and Equation 1, the generating process is formulated as:

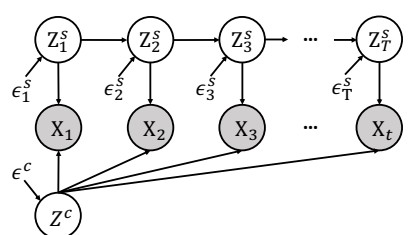

$$\begin{cases} \mathbf{x}_t = g(\mathbf{z}_t^s, \mathbf{z}^c), \\ z_{t,i}^s = f_i^s \left( \mathbf{Pa}(z_{t,i}^s), \epsilon_{t,i}^s \right), \text{ with } \begin{cases} \epsilon_{t,i}^s \sim p_{\epsilon_i}^s, \\ \epsilon_j^c \sim p_{\epsilon_j}^c, \end{cases} \\ z_j^c = f_j^c \left( \epsilon_j^c \right), \end{cases} \quad (1)$$

in which $z_{t,i}^s, z_j^c \in \mathbb{R}$ refers to the $i$-th entry of $\mathbf{z}_t^s$ and $j$-th entry of $\mathbf{z}^c$, respectively. $\mathbf{Pa}(z_{t,i}^s)$ refers to the time-delayed parents of $z_{t,i}^s$. All noise terms $\epsilon_{t,i}$ and $\epsilon_j$ are independently sampled from their respective distributions: $p_{\epsilon_i}^s$ for the $i$-th entry of the style dynamics, and $p_{\epsilon_j}^c$ for the $j$-th entry of the content elements. The components of $\mathbf{z}_t^s$

Figure 2: **The generating process.** The gray shade of nodes indicates that the variable is observable.

are mutually independent, conditioned on all historical variables $\cup_{i=1}^{n_s} \mathbf{Pa}(z_{t,i}^s)$. The non-parametric causal transition $f_i$ enables an arbitrarily nonlinear interaction between the noise term $\epsilon_{t,i}$ and the set of parent variables $\mathbf{Pa}(z_{t,i}^s)$, allowing for flexible modeling of the style dynamics.

### 2.2 IDENTIFICATION OF THE LATENT CAUSAL PROCESS

For simplicity, we denote $\mathbf{f}^s$ as the group of functions $\{f_i^s\}_{i=1}^{n_s}$, and similarly for $\mathbf{f}^c, \mathbf{p}^s, \mathbf{p}^c$.

**Definition 2.1** (Observational Equivalence). Let $V = \{\mathbf{x}_1, \mathbf{x}_2, \cdots, \mathbf{x}_T\}$ represent the observed video generated by the true generating process specified by $(g, \mathbf{f}^s, \mathbf{f}^c, \mathbf{p}^s, \mathbf{p}^c)$, as defined in Equation 1. A learned model $(\hat{g}, \hat{\mathbf{f}}^s, \hat{\mathbf{f}}^c, \hat{\mathbf{p}}^s, \hat{\mathbf{p}}^c)$ is observationally equivalent to the true process if the model distribution $p_{(\hat{g}, \hat{\mathbf{f}}^s, \hat{\mathbf{f}}^c, \hat{\mathbf{p}}^s, \hat{\mathbf{p}}^c)}(V)$ matches the data distribution $p_{(g, \mathbf{f}^s, \mathbf{f}^c, \mathbf{p}^s, \mathbf{p}^c)}(V)$ for all values of $V$.

**Illustration.** When observational equivalence is achieved, the distribution of videos generated by the model exactly matches that of the ground truth, i.e., the training set. In other words, the model produces video data that is indistinguishable from the actual observed data.

**Definition 2.2** (Block-wise Identification of Generating Process). Let the true generating process be $(g, \mathbf{f}^s, \mathbf{f}^c, \mathbf{p}^s, \mathbf{p}^c)$ as specified in Equation 1 and let its estimation be $(\hat{g}, \hat{\mathbf{f}}^s, \hat{\mathbf{f}}^c, \hat{\mathbf{p}}^s, \hat{\mathbf{p}}^c)$. The generating process is identifiable up to the subspace of style dynamics and content elements, if the observational equivalence ensures that the estimated $(\hat{\mathbf{z}}_t^s, \hat{\mathbf{z}}^c)$ satisfies the condition that there exist bijective mappings from $(\hat{\mathbf{z}}_t^s, \hat{\mathbf{z}}^c)$ to $(\mathbf{z}_t^s, \mathbf{z}^c)$ and from $\hat{\mathbf{z}}^c$ to $\mathbf{z}^c$. Formally, there exists invertible functions $h : \mathbb{R}^{n_s + n_c} \to \mathbb{R}^{n_s + n_c}$ and $h_c : \mathbb{R}^{n_c} \to \mathbb{R}^{n_c}$ such that

$$p_{(\hat{g}, \hat{\mathbf{f}}^s, \hat{\mathbf{f}}^c, \hat{\mathbf{p}}^s, \hat{\mathbf{p}}^c)}(V) = p_{(g, \mathbf{f}^s, \mathbf{f}^c, \mathbf{p}^s, \mathbf{p}^c)}(V) \Rightarrow \quad [\mathbf{z}_t^s, \mathbf{z}^c] = h([\hat{\mathbf{z}}_t^s, \hat{\mathbf{z}}^c]), \mathbf{z}^c = h_c(\hat{\mathbf{z}}^c), \quad (2)$$

where $[\cdot]$ denotes concatenation.

**Definition 2.3** (Component-wise Identification of Style Dynamics). On top of Definition 2.2, when $h^s$ is a combination of permutation $\pi$ and a component-wise invertible transformation $\mathcal{T}$. Formally,

$$p_{(\hat{g}, \hat{\mathbf{f}}^s, \hat{\mathbf{f}}^c, \hat{\mathbf{p}}^s, \hat{\mathbf{p}}^c)}(V) = p_{(g, \mathbf{f}^s, \mathbf{f}^c, \mathbf{p}^s, \mathbf{p}^c)}(V) \Rightarrow \quad \mathbf{z}_t^s = (\pi \cdot \mathcal{T})(\hat{g}_s^{-1}(\mathbf{x}_t)). \quad (3)$$

## 2.3 FROM IDENTIFIABILITY TO CONTROLLABLE VIDEO GENERATION

Identifiability ensures the uniqueness of the latent representation, meaning that the learned latent variables correspond to the true latent variables up to certain allowable transformations. Moreover, identifiability can be defined at different levels, with higher levels indicating that the estimated variables align more closely with the true underlying factors. Stronger identifiability thus leads to more disentangled representations, enabling more efficient and precise control over video generation.

When block-wise identifiability is achieved, content-related components are effectively disentangled from motion. This allows motion control to be applied independently, enabling manipulation of motion without altering the underlying content of the video. For example, in a sequence where a camera moves forward, one can adjust the camera's direction without affecting the static scene.

In contrast, when component-wise identifiability is achieved, each learned component corresponds one-to-one with its true underlying factor. This property allows independent manipulation of each generative factor by adjusting the corresponding latent dimension, without undesired interference across factors. Such fine-grained control represents the ultimate goal of disentanglement, providing a principled and theoretically grounded formulation. For instance, in generating a video of a face, one can separately adjust head movements or eye blinks by modifying the associated latent variables.

The key remaining question, therefore, is under what conditions identifiability can be guaranteed, and how such guarantees can be established in practice.

## 3 THEORETICAL ANALYSIS

In this section, we discuss the conditions under which the block-wise identification (Definition 2.2) and component-wise identification (Definition 2.3) hold. We consider only stationary latent process in this part for simplicity. However, our theorem can be extended to nonstationary scenarios when supervision is provided, as discussed in Appendix A.4.

### 3.1 BLOCK-WISE IDENTIFICATION

Without loss of generality, we first consider the case where $\mathrm{Pa}(z_{t,i}^s) = \mathbf{z}_{t-1}^s$, meaning that the time-dependent effects are governed by the dynamics of the previous time step.

**Definition 3.1.** (Linear Operator(Hu & Schennach, 2008a; Dunford & Schwartz, 1988)) Consider two random variables $a$ and $b$ with support $\mathcal{A}$ and $\mathcal{B}$, the linear operator $L_{b|a}$ is defined as a mapping from a density function $p_a$ in some function space $\mathcal{F}(\mathcal{A})$ onto the density function $L_{b|a} \circ p_a$ in some function space $\mathcal{F}(\mathcal{B})$,

$$\mathcal{F}(\mathcal{A}) \to \mathcal{F}(\mathcal{B}): \ p_b = L_{b|a} \circ p_a = \int_{\mathcal{A}} p_{b|a}(\cdot \mid a) p_a(a) da.$$

To better illustrate Linear Operator, examples of linear operators are provided in the Appendix A.3.

**Theorem 3.2** (Block-wise Identifiability). *Consider video observation $V = \{\mathbf{x}_1, \mathbf{x}_2, \ldots, \mathbf{x}_T\}$ generated by process $(g, \boldsymbol{f}^s, \boldsymbol{f}^c, \boldsymbol{p}^s, \boldsymbol{p}^c)$ with latent variables denoted as $\mathbf{z}_t^s$ and $\mathbf{z}_c$, according to Equation 1, where $\mathbf{x}_t \in \mathbb{R}^{n_x}, \mathbf{z}_t^s \in \mathbb{R}^{n_s}, \mathbf{z}^c \in \mathbb{R}^{n_c}$. If assumptions*

- *B1 (Positive Density) the probability density function of latent variables is always positive and bounded;*

- *B2 (Minimal Changes) the linear operators $L_{\mathbf{x}_{t+1}|\mathbf{z}_t^s, \mathbf{z}^c}$ and $L_{\mathbf{x}_{t-1}|\mathbf{x}_{t+1}}$ are injective for a bounded function space;*

- *B3 (Weakly Monotonic) for any $\dot{\mathbf{z}}_t, \ddot{\mathbf{z}}_t \in \mathcal{Z}^c \times \mathcal{Z}_t^s$ ($\dot{\mathbf{z}}_t \neq \ddot{\mathbf{z}}_t$), the set $\{\mathbf{x}_t : p(\mathbf{x}_t|\dot{\mathbf{z}}_t) \neq p(\mathbf{x}_t|\ddot{\mathbf{z}}_t)\}$ has positive probability, and conditional densities are bounded and continuous;*

*are satisfied, then $\mathbf{z}_t$ is block-wise identifiable with regard to $\hat{\mathbf{z}}_t$ from learned model $(\hat{g}, \hat{\boldsymbol{f}}^s, \hat{\boldsymbol{f}}^c, \hat{\boldsymbol{p}}^s, \hat{\boldsymbol{p}}^c)$ under Observation Equivalence.*

**Illustration of assumptions.** The assumptions above are commonly used in the literature on the identification of latent variables under measurement error (Hu & Schennach, 2008b). Firstly, Assumption B1 requires a continuous distribution. Secondly, Assumption B2 imposes a minimal requirement on the number of variables. The linear operator $L_{b|a}$ ensures that there is sufficient variation in the density of $b$ for different values of $a$, thereby guaranteeing injectivity. In a video, $\mathbf{x}_t$ is of much higher dimensionality compared to the latent variables. As a result, the injectivity assumption is easily satisfied. In practice, following the principle of minimal changes, if a model with fewer latent variables can successfully achieve observational equivalence, it is more likely to learn the true distribution. Assumption B3 requires the distribution of $\mathbf{x}_t$ changes when the value of latent variables changes. This assumption is much weaker compared to the widely used invertibility assumption adopted by previous works, such as (Yao et al., 2022).

Overall, the three assumptions impose mild requirements on the underlying data generation process. When these assumptions are satisfied, the block-wise identifiability result established in our theorem holds. Importantly, real-world video scenarios naturally conform to these assumptions, and the structure of many existing generative models is already aligned with the conditions specified in the theorem. A more detailed discussion of these assumptions and their practical relevance can be found in Appendix A.1.

**Proof sketch.** We separately prove the identifiability of all latent variables and $\mathbf{z}^c$. For the first part, it is built on (Fu et al., 2025), following the line of work from (Hu & Schennach, 2008b). Intuitively, it demonstrates that a minimum of 3 different observations of latent variables are required for identification under the given data generation process. For the second part, we use the contradiction to show that the same $\mathbf{z}^c$ in different frames of a video can be identified leveraging the invariance.

## 3.2 Component-wise Identification

**Theorem 3.3** (Component-wise Identifiability). *Consider video observation $V = \{\mathbf{x}_1, \mathbf{x}_2, \ldots, \mathbf{x}_T\}$ generated by process $(g, \boldsymbol{f^s}, \boldsymbol{f^c}, \boldsymbol{p^s}, \boldsymbol{p^c})$ with latent variables denoted as $\mathbf{z}_t^s$ and $\mathbf{z}_c$, according to Equation 1, where $\mathbf{x}_t \in \mathbb{R}^{n_x}, \mathbf{z}_t^s \in \mathbb{R}^{n_s}, \mathbf{z}^c \in \mathbb{R}^{n_c}$. Suppose assumptions in Theorem 3.2 hold. If assumptions*

- *C1 (Smooth and Positive Density) the probability density function of latent variables is always third-order differentiable and positive;*

- *C2 (Sufficient Changes) let $\eta_{t,i} \triangleq \log p(z_{t,i}^s | \mathbf{z}_{t-1}^s)$ and*

$$\mathbf{v}_{t,l} \triangleq \left( \frac{\partial^2 \eta_{t,1}}{\partial z_{t,1} \partial z_{t-1,l}}, \cdots, \frac{\partial^2 \eta_{t,n_s}}{\partial z_{t,n} \partial z_{t-1,l}} \right) \oplus \left( \frac{\partial^3 \eta_{t,1}}{\partial^2 z_{t,1} \partial z_{t-1,l}}, \cdots, \frac{\partial^3 \eta_{t,n_s}}{\partial^2 z_{t,n} \partial z_{t-1,l}} \right), \qquad (4)$$

  *for $l \in \{1, 2, \cdots, n\}$. For each value of $\mathbf{z}_t$, there exists $2n_s$ different values of $z_{t-1,l}$ such that the $2n_s$ vectors $\mathbf{v}_{t,l} \in \mathbb{R}^{2n_s}$ are linearly independent;*

- *C3 (Conditional Independence) the learned $\hat{\mathbf{z}}_t^s$ is independent with $\hat{\mathbf{z}}^c$, and all entries of $\hat{\mathbf{z}}_t^s$ are mutually independent conditioned on $\hat{\mathbf{z}}_{t-1}^s$;*

*are satisfied, then $\mathbf{z}_t^s$ is component-wise identifiable with regard to $\hat{\mathbf{z}}_t^s$ from learned model $(\hat{g}, \hat{\boldsymbol{f}}^s, \hat{\boldsymbol{f}}^c, \hat{\boldsymbol{p}}^s, \hat{\boldsymbol{p}}^c)$ under Observation Equivalence.*

**Proof sketch.** In summary, component-wise identification relies on the changeability of style dynamics, i.e., sufficient changes. Starting from the results of block-wise identification, we establish the connection between $\mathbf{z}_t^s$ and $\hat{\mathbf{z}}_t^s$ in terms of their distributions, i.e., $p(\mathbf{z}_t) = p(\hat{\mathbf{z}}_t) \cdot |H_t|$, where $H_t$ is the jacobian matrix. Leveraging the second-order derivative of the log probability, we construct a system of equations with terms of $\frac{\partial z_{t,i}^s}{\partial \hat{z}_{t,j}^s} \cdot \frac{\partial z_{t,i}^s}{\partial \hat{z}_{t,k}^s}$ and coefficients as specified in assumption C2. We leverage the third-order derivative of the previous latent variable $z_{t-1,l}$ to eliminate $|H_t|$, utilizing the fact that the history does not influence the current mapping from estimation to truth. Solving this system yields $\frac{\partial z_{t,i}^s}{\partial \hat{z}_{t,j}^s} \cdot \frac{\partial z_{t,i}^s}{\partial \hat{z}_{t,k}^s} = 0$. This indicates that $z_{t,i}^s$ is a function of at most one $\hat{z}_{t,j}^s$.

**Illustration of assumptions.** The assumptions C1, C2 on the data generating process are commonly adopted in existing identifiable results for Temporally Causal Representation Learning (Yao et al., 2022). Specifically, C1 implies that the latent variables evolve continuously over time, while C2

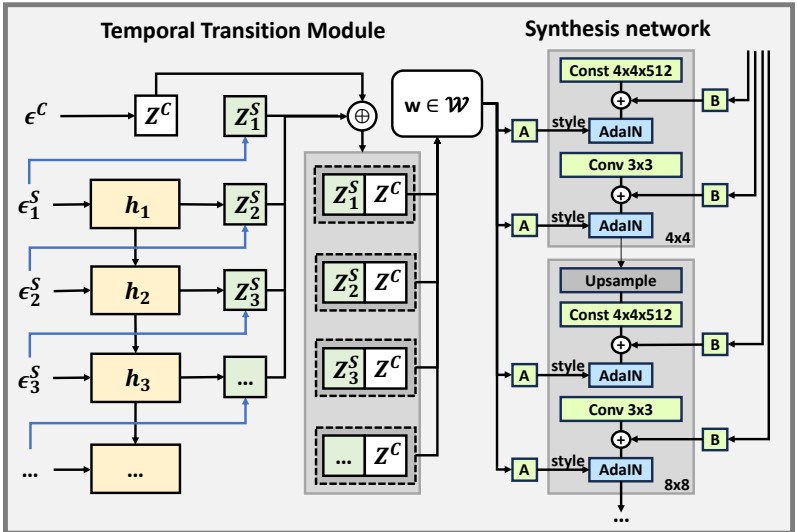

Figure 3: **Generator** operates from left to right, beginning with a random noise input. The noise first passes through a Temporal Transition Module, which produces a disentangled representation of the underlying factors. This representation is then fed into the synthesis network to generate frames at the pixel level. In the figure, the blue arrow illustrates the Deep Sigmoid Flow.

ensures that the variability in the data can be effectively captured. The assumption C3 constraints to the learned model require it to separate different variables of style dynamics into independent parts.

## 4 APPROACH

Given our results on identifiability, we implement our model, CoVoGAN. Our architecture is based on StyleGAN2-ADA (Karras et al., 2020), incorporating a Temporal Transition Module in the Generator to enforce the minimal change principle and sufficient change property. Additionally, we add a Video Discriminator to ensure observational equivalence of the joint distribution $p(V)$.

### 4.1 MODEL STRUCTURE

**Noise sampling.** The structure of the generator is shown in Figure 3. To generate a video with length $T$, we first independently sample random noise from a normal distribution $\epsilon \sim \mathcal{N}(\mathbf{0}, \mathbf{I})$. We then naively split it into several parts, i.e., $\epsilon = [\epsilon^c; \epsilon_1^s; \epsilon_2^s; \cdots; \epsilon_T^s]$.

**Temporal Transition Module.** Following the generating process in Equation 1, we handle $\mathbf{z}^c$ and $\mathbf{z}_t^s$ separately. On the one hand, we employ an autoregressive model to capture historical information, followed by a conditional flow to generate $\mathbf{z}_t^s$. Specifically, we implement a Gated Recurrent Unit (GRU) (Chung et al., 2014) and Deep Sigmoid Flow (DSF) (Huang et al., 2018), formulated as

$$\mathbf{h}_t = \text{GRU}(\mathbf{h}_{t-1}, \epsilon_{t-1}^s), z_{t,i}^s = \text{DSF}_i(\epsilon_{t,i}^s; \mathbf{h}_{t-1}), \tag{5}$$

where $\mathbf{h}_t$ denotes all the historical information until time step $t$, and $z_{t,i}^s$ will be mutually independent conditioned on $\mathbf{h}_t$. On the other hand, since $\mathbf{z}^c$ are not required to be mutually independent, we use an MLP to generate $\mathbf{z}^c$, i.e.,

$$\mathbf{z}^c = \text{MLP}(\epsilon^c). \tag{6}$$

Concatenate $\mathbf{z}_t^s$ and $\mathbf{z}^c$, and then we obtain the disentangled representation $\mathbf{z}_t = \mathbf{z}^c \oplus \mathbf{z}_t^s$ for each frame at time step $t$ of the video.

**Synthesis network.** The synthesis network is designed in the same way as StyleGAN2-ADA. The generated representation $\mathbf{z}_t$ is first fed into the mapping network to obtain a semantic vector $w(\mathbf{z}_t) \in \mathcal{W}$, and then the $t$-th frame of the video is generated by the convolutional network with $w$.

**Discriminator structure.** To ensure observational equivalence, we implement a video discriminator $D_V$ separate from the image discriminator $D_I$. For the image discriminator, we follow the design of

the original StyleGAN2-ADA. For the video discriminator, we adopt a channel-wise concatenation of activations at different resolutions to model and manage the spatiotemporal output of the generator.

**Loss.** In addition to the original loss function of StyleGAN2-ADA, we introduce two additional losses: (1) a video discriminator loss, and (2) a mutual information maximization term (Chen et al., 2016) between the latent dynamic variables $\mathbf{z}_t^s$ and the intermediate layer outputs of the video discriminator. This encourages the model to learn a more informative and structured representation.

## 4.2 RELATIONSHIP BETWEEN MODEL AND THEOREM.

**Block-wise identification.** As discussed in Theorem 3.2, achieving block-wise identifiability benefits from minimizing the dimension $n_s$ of the style dynamics, especially when the true $n_s$ is unknown. In practice, this translates to a hyperparameter selection question. Therefore, we opted for a relatively modest value of $n_s$ and observed that it suffices to attain a satisfactory level of disentanglement capability. Furthermore, as required by the assumptions, the learned variables $\hat{\mathbf{z}}_t^s$ and $\hat{\mathbf{z}}^c$ are block-wise independent, i.e., $\hat{\mathbf{z}}_t^s \perp\!\!\!\perp \hat{\mathbf{z}}^c$. This independence is necessary to achieve block-wise identifiability.

**Component-wise identification.** As outlined in Theorem 3.3, the sufficient change property is a critical assumption for achieving identifiability. To enforce temporally conditional independence, we employ a component-wise flow model that transforms a set of independent noise variables $\epsilon_t^s$ into the style dynamics, conditioned on historical information. Furthermore, the flow model is designed to maximally preserve the information from $\epsilon_{t,i}^s$ to $z_{t,i}^s$, enabling the model to effectively capture sufficient variability in the data. Note that when computing the historical information $h_t$, we utilize $\epsilon_t^s$ instead of $\mathbf{z}_t^s$ (as illustrated in the generating process in Equation 1) as the condition for the component-wise flow. This approach offers two key advantages. First, it simplifies the model architecture since the flow does not need to incorporate the output from another flow. Second, the noise terms already fully characterize the corresponding style dynamics, which remains consistent with the theoretical framework. Furthermore, given that the precise time lag of dynamic variables remains unspecified a priori in the dataset, the GRU's gating mechanism can selectively filter out irrelevant historical information that lies outside $\text{Pa}(\mathbf{z}_t)$. This capability enables the model to demonstrate significantly superior performance compared to traditional non-gated architectures, such as vanilla RNNs. A detailed ablation study is presented in Section 5.4.

# 5 EXPERIMENTS

## 5.1 EXPERIMENTAL SETUP

**Datasets.** We evaluate our model on four different real-world datasets: FaceForensics (Rössler et al., 2018), SkyTimelapse (Xiong et al., 2018), RealEstate (Zhou et al., 2018) and CelebV-HQ (Zhu et al., 2022). The first three datasets contain videos with a resolution of $256 \times 256$ pixels, while the last dataset consists of videos at $512 \times 512$ resolution. We employ standard train-test splits for fair evaluation. Detailed information about the datasets are provided in Appendix C.

**Evaluation metrics.** To comprehensively evaluate the performance of CoVoGAN, we employ both quantitative and qualitative assessment metrics. For quantitative evaluation, we adopt the Fréchet Video Distance (FVD) (Unterthiner et al., 2018), a widely-used metric for assessing video generation quality. We report FVD scores at two different temporal scales: $\text{FVD}_8$ and $\text{FVD}_{16}$, where the subscript denotes the number of frames in a video. To better assess the disentanglement capability of our method, we compare the widely used disentanglement metrics SAP (Kumar et al., 2017) and modularity (Ridgeway & Mozer, 2018). Additionally, we measure the Mean Correlation Coefficient (MCC), a standard metric for disentanglement. Details of the metrics can be found in Appendix D.

## 5.2 VIDEO QUALITY

We consider four GAN-based models: MoCoGAN-HD, DIGAN, StyleGAN-V, and MoStGAN-V. We also compare three diffusion-based models: VDM (Ho et al., 2022), LVDM (He et al., 2022) and Latte (Ma et al., 2024). For MoCoGAN-HD, we freeze the image generator pretrained by the original authors and train the motion generator and discriminator. We exclude MoCoGAN-HD from the comparisons on the specific datasets without pretrained image generator. For Latte, we directly use the released checkpoints. For the remaining baselines, we train the models from scratch using their official implementations to ensure comparability. The quantitative results in Table **??** show

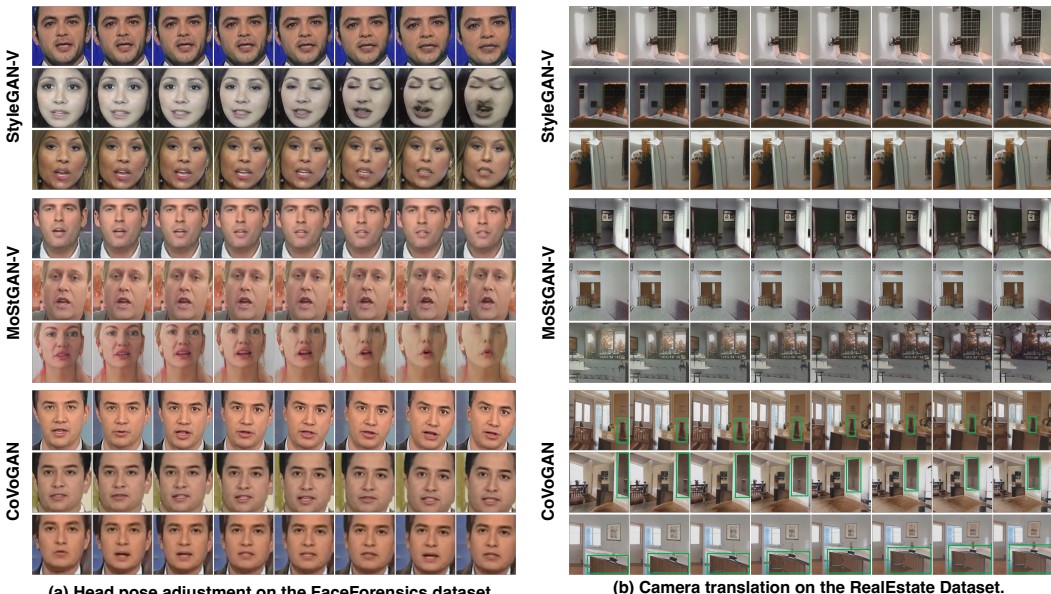

(a) Head pose adjustment on the FaceForensics dataset.

(b) Camera translation on the RealEstate Dataset.

Figure 4: Controllability in the latent space across datasets and methods. Each method is evaluated with three samples by varying a single latent dimension. Only CoVoGAN exhibits consistent control across identities: (a) head pose adjustment on FaceForensics, (b) camera translation on RealEstate.

| Method | FaceForensics | | SkyTimelapse | | RealEstate | | CelebV-HQ |
|---|---|---|---|---|---|---|---|
| | $FVD_8$ | $FVD_{16}$ | $FVD_8$ | $FVD_{16}$ | $FVD_8$ | $FVD_{16}$ | $FVD_{16}$ |
| MoCoGAN-HD | 140.05 | 185.51 | 1214.13 | 1721.89 | – | – | 412.50 |
| DIGAN | 57.52 | 61.65 | 60.54 | 105.03 | 182.86 | 178.27 | – |
| StyleGAN-V | 49.24 | 52.70 | 45.30 | 62.55 | 199.66 | 201.95 | 147.81 |
| MoStGAN-V | 47.67 | 49.85 | 40.97 | 55.36 | 247.77 | 265.54 | 127.62 |
| VDM | 1038.29 | 1046.60 | 1099.21 | 1104.80 | 1524.17 | 1526.04 | – |
| LVDM | 136.60 | 153.38 | 307.22 | 319.67 | 423.54 | 448.31 | – |
| Latte | 45.49 | 49.02 | 40.21 | **41.84** | – | – | – |
| **CoVoGAN** | **43.75** | **48.80** | **35.58** | 46.51 | **154.88** | **174.87** | **97.16** |

Table 1: $FVD_8 \downarrow$ and $FVD_{16} \downarrow$ results across different datasets.

that CoVoGAN consistently achieves top performance across all datasets, despite diffusion-based models occasionally exhibiting higher visual fidelity. More visualization results are provided in the Appendix F.5.

## 5.3 CONTROLLABILITY

**Block-wise disentanglement.** Figure 4 demonstrates the video controllability of CoVoGAN in comparison with baseline methods, highlighting our model's superior capability for disentanglement between motion and content. The analysis follows a systematic procedure: we first generate a base video sequence, then apply a controlled modification by adding a value to specific motion-related latent variables.

For CoVoGAN, we modify one dimension of the style dynamics $z_t^s$. For StyleGAN-V and MoStGAN-V, we manipulate one

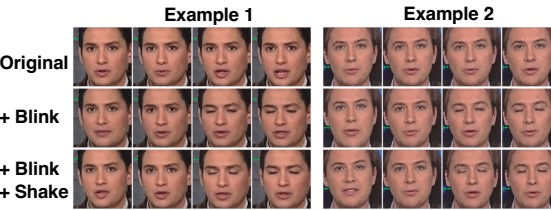

Figure 5: Controllability visualization results on the Face-Forensics dataset. Two distinct motion concepts are manipulated to illustrate component-wise disentanglement. Corresponding videos are provided in the supplementary materials for better visualization.

dimension of their (latent) motion code. We apply equivalent modifications to the corresponding latent dimensions in each baseline model. To validate the consistency of our controllability analysis, we randomly sample three distinct video sequences and apply identical modifications to their respective

latent representations. Other baselines are not compared since there are no specific variables with semantic information.

The results show that our proposed CoVoGAN model learns a disentangled representation that effectively separates style dynamics from content elements. (1) This disentanglement enables independent manipulation of motion characteristics while preserving content consistency. (2) A key advantage of this approach is that identical modifications to the style latent space consistently produce similar motion patterns across different content identities. Baseline models achieve only partial disentanglement, exhibiting two major limitations: (1) visual distortions of the modified videos and (2) inconsistent or misaligned motion patterns when applied to different identities.

**Component-wise disentanglement.** We also show the precise control over individual motion components. This capability is enabled by the component-wise identifiability of our model, which ensures each latent dimension corresponds to a specific and interpretable motion attribute, e.g., eye blinking or

Table 2: Disentanglement Comparison across different models.

| Metrics | StyleGAN-V | MoStGAN-V | LVDM | Latte | CoVoGAN |
|---|---|---|---|---|---|
| MCC (%) ↑ | 29.00 | 27.95 | 21.60 | 20.87 | **33.78** |
| SAP (%) ↑ | 4.25 | 5.90 | 0.72 | 0.75 | **8.48** |
| Modularity (%) ↑ | 7.66 | 13.48 | 7.25 | 7.44 | **17.37** |

head shaking. Our experimental procedure begins with randomly sampling two distinct video sequences, as illustrated in the first row of Figure 5. We then selectively modify the latent dimension corresponding to eye blinking dynamics in the second line. Subsequently, we modify a second latent dimension controlling head shaking motion while maintaining the previously adjusted eye blinking pattern in the last line. The results show naturalistic head movements from left to right, synchronized with the preserved eye blinking, illustrating our model's capability for independent yet coordinated control of multiple motion components.

**Disentanglement metrics.** We first extract semantic annotations from the videos for disentanglement evaluation. For this purpose, we conduct experiments on the FaceForensics dataset, which facilitates the extraction of meaningful facial attributes. We utilize a pretrained model from Dlib to extract facial landmarks and compute semantic annotations, including eye size, mouth size, head position, and head angle. We only compare our method with two models that explicitly incorporate semantic representation layers, which we use to compute the metrics. Since diffusion-based models do not provide a compact latent representation by design, we adopt the following procedure: we first randomly sample videos, then extract frame-wise representations from the high-dimensional latent space. These representations are reduced to 128 dimensions using PCA, and the disentanglement metrics are computed in this reduced space. For our method, we compute the metrics using the dynamic latent variables $z_t^s$. As shown in Table 2, our method achieves the best performance.

## 5.4 ABLATION STUDY

We conduct an ablation study to evaluate the contributions of the proposed components within the Temporal Transition Module, as shown in Table 3. Replacing the GRU with a standard RNN leads to a noticeable performance drop. This is primarily due to the loss of sparsity provided by the GRU's gating mechanism, which helps isolate time-delayed effects. Without this mecha-

Table 3: Ablation studies on different GRU configurations and component-wise flow on the FaceForensics dataset.

| Metric | CoVoGAN | w/o GRU | w/o flow |
|---|---|---|---|
| FVD$_{16}$ ↓ | **48.80** | 53.68 | 82.81 |
| MCC (%) ↑ | **33.78** | 26.59 | 8.22 |
| SAP (%) ↑ | **8.48** | 7.25 | 0.55 |
| Modularity (%) ↑ | **17.37** | 12.40 | 10.24 |

nism, the model faces a larger search space, making it harder to learn an effective transition function. Substituting the component-wise flow with a fully connected MLP results in an even more significant degradation. This decline can be attributed to two factors: (1) the mutual independence between style dynamics can no longer be maintained, and (2) capturing sufficient changes becomes more difficult.

## 6 CONCLUSION

In this paper, we proposed a Temporal Transition Module and implemented it in a GAN to achieve CoVoGAN. By leveraging the principles of minimal and sufficient changes, we successfully disentangled (1) the motion and content, and (2) different concepts within the motion. We established an identifiability guarantee for both block-wise and component-wise disentanglement. Our proposed CoVoGAN model demonstrates high generative quality and controllability. We validated the performance on various datasets and conducted ablation experiments to further confirm the effectiveness of our model. Overall, our work provides a principled and practical framework for disentangled video generation and opens new directions for fine-grained, interpretable, and controllable visual synthesis. **Limitations and Future works:** It focuses primarily on theoretical contributions and their empirical validation, while the integration with higher-fidelity generative architectures and application to open-domain scenarios remains an important direction for future work.

## 7 ACKNOWLEGEMENT

We would also like to acknowledge the support from NSF Award No. 2229881, AI Institute for Societal Decision Making (AI-SDM), the National Institutes of Health (NIH) under Contract R01HL159805, and grants from Quris AI, Florin Court Capital, MBZUAI-WIS Joint Program, and the Al Deira Causal Education project.

## REPRODUCIBILITY STATEMENT

We have taken several steps to ensure the reproducibility of our results. A complete description of the datasets and preprocessing steps is included in Appendix C. The implementation details, including model architectures, training hyperparameters, and optimization settings, are provided in Appendix E. Extended results are reported in Appendix F. In addition, we release the full source code and configuration files in the supplementary materials.

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

*Appendix for*

# "Controllable Video Generation with Provable Disentanglement"

## A    IDENTIFIABILITY THEORY

### A.1    PROOF

Without loss of generality, we first consider the case where $\mathrm{Pa}(z^s_{t,i}) = \mathbf{z}^s_{t-1}$, meaning that the time-dependent effects are governed by the dynamics of the previous time step.

**Theorem A1** (Blockwise Identifiability). *Consider video observation $V = \{\mathbf{x}_1, \mathbf{x}_2, \ldots, \mathbf{x}_T\}$ generated by process $(g, \boldsymbol{f}^s, \boldsymbol{f}^c, \boldsymbol{p}^s, \boldsymbol{p}^c)$ with latent variables denoted as $\mathbf{z}^s_t$ and $\mathbf{z}_c$, according to Equation 1, where $\mathbf{x}_t \in \mathbb{R}^{n_x}, \mathbf{z}^s_t \in \mathbb{R}^{n_s}, \mathbf{z}^c \in \mathbb{R}^{n_c}$. If assumptions*

- *B1 (Positive Density) the probability density function of latent variables is always positive and bounded;*

- *B2 (Minimal Changes) the linear operators $L_{\mathbf{x}_{t+1}|\mathbf{z}^s_t, \mathbf{z}^c}$ and $L_{\mathbf{x}_{t-1}|\mathbf{x}_{t+1}}$ are injective for bounded function space;*

- *B3 (Weakly Monotonic) for any $\dot{\mathbf{z}}_t, \ddot{\mathbf{z}}_t \in \mathcal{Z}^c \times \mathcal{Z}^s_t$ $(\dot{\mathbf{z}}_t \neq \ddot{\mathbf{z}}_t)$, the set $\{\mathbf{x}_t : p(\mathbf{x}_t|\dot{\mathbf{z}}_t) \neq p(\mathbf{x}_t|\ddot{\mathbf{z}}_t)\}$ has positive probability, and conditional densities are bounded and continuous;*

*are satisfied, then $\mathbf{z}_t$ is blockwisely identifiable with regard to $\hat{\mathbf{z}}_t$ from learned model $(\hat{g}, \hat{\boldsymbol{f}}^s, \hat{\boldsymbol{f}}^c, \hat{\boldsymbol{p}}^s, \hat{\boldsymbol{p}}^c)$ under Observation Equivalence.*

*Proof.* We first prove the monoblock identification of $\mathbf{z}^s_t, \mathbf{z}^c = h(\hat{\mathbf{z}}^s_t, \hat{\mathbf{z}}^c)$, then we prove the blockwise identification $\mathbf{z}^c = h_c(\hat{\mathbf{z}}^c)$.

**Monoblock Identification.** Following (Hu & Shum, 2012; Hu & Schennach, 2008b), when assumptions B1, B2, B3 satisfied, the blockwise identifiability of $[\mathbf{z}^s_t, \mathbf{z}^c]$ is assured, according to Theorem 3.2 (Monoblock identifiability) in (Fu et al., 2025). In short ,there exists an invertible function $g$ such that $[\mathbf{z}^s_t, \mathbf{z}^c] = h(\hat{\mathbf{z}}^s_t, \hat{\mathbf{z}}^c)$, where $[\cdot]$ denotes concatenation.

**Identification of $\mathbf{z}^c$.** We prove this by contradiction. Suppose that for any $\hat{\mathbf{z}}^c$, we have

$$\mathbf{z}^c = h_c(\hat{\mathbf{z}}^s_t, \hat{\mathbf{z}}^c), \tag{7}$$

where there exist at least two distinct values of $\hat{\mathbf{z}}^s_t$ such that $\mathbf{z}^c$ takes different values.

When observational equivalence holds, the function remains the same for all values of $t$:

$$h_c(\hat{\mathbf{z}}^s_t, \hat{\mathbf{z}}^c) = g_c^{-1}(\mathbf{x}_t) = (g_c^{-1} \circ \hat{g})(\hat{\mathbf{z}}^s_t, \hat{\mathbf{z}}^c), \tag{8}$$

where $g_c^{-1}$ first demix $\mathbf{x}_t$ then extract the content part, as defined in Equation 2.

For any two distinct $t \neq t'$, we have

$$h_c(\hat{\mathbf{z}}^s_t, \hat{\mathbf{z}}^c) = \mathbf{z}^c = h_c(\hat{\mathbf{z}}^s_{t'}, \hat{\mathbf{z}}^c), \tag{9}$$

which holds for all pairs $(\hat{\mathbf{z}}^s_t, \hat{\mathbf{z}}^s_{t'})$ within the domain of definition.

According to Assumption B1, the joint distribution $p(\hat{\mathbf{z}}^s_t, \hat{\mathbf{z}}^s_{t'})$ is always positive. Thus, to satisfy Equation 9, $\hat{\mathbf{z}}^s_t$ must not contribute to $\mathbf{z}^c$ through $h_c$. In other words, we obtain

$$\mathbf{z}^c = h_c(\hat{\mathbf{z}}^c). \tag{10}$$

$\square$

**Theorem A2** (Component-wise Identifiability). *Consider video observation $V = \{\mathbf{x}_1, \mathbf{x}_2, \ldots, \mathbf{x}_T\}$ generated by process $(g, \boldsymbol{f}^s, \boldsymbol{f}^c, \boldsymbol{p}^s, \boldsymbol{p}^c)$ with latent variables denoted as $\mathbf{z}^s_t$ and $\mathbf{z}_c$, according to Equation 1, where $\mathbf{x}_t \in \mathbb{R}^{n_x}, \mathbf{z}^s_t \in \mathbb{R}^{n_s}, \mathbf{z}^c \in \mathbb{R}^{n_c}$. Suppose assumptions in Theorem A1 hold. If assumptions*

- **C1 (Smooth and Positive Density)** the probability density function of latent variables is always third-order differentiable and positive;

- **C2 (Sufficient Changes)** let $\eta_{t,i} \triangleq \log p(z_{t,i}^s | \mathbf{z}_{t-1}^s)$ and

$$\mathbf{v}_{t,l} \triangleq \Big( \frac{\partial^2 \eta_{t,1}}{\partial z_{t,1} \partial z_{t-1,l}}, \cdots, \frac{\partial^2 \eta_{t,n_s}}{\partial z_{t,n} \partial z_{t-1,l}} \Big) \oplus \Big( \frac{\partial^3 \eta_{t,1}}{\partial^2 z_{t,1} \partial z_{t-1,l}}, \cdots, \frac{\partial^3 \eta_{t,n_s}}{\partial^2 z_{t,n} \partial z_{t-1,l}} \Big), \quad (11)$$

for $l \in \{1, 2, \cdots, n\}$. For each value of $\mathbf{z}_t$, there exists $2n_s$ different of values of $z_{t-1,l}$ such that the $2n_s$ vector $\mathbf{v}_{t,l} \in \mathbb{R}^{2n_s}$ are linearly independent;

- **C3 (Conditional Independence)** the learned $\hat{\mathbf{z}}_t^s$ is independent with $\hat{\mathbf{z}}^c$, and all entries of $\hat{\mathbf{z}}_t^s$ are mutually independent conditioned on $\hat{\mathbf{z}}_{t-1}^s$;

are satisfied, then $\mathbf{z}_t^s$ is component-wisely identifiable with regard to $\hat{\mathbf{z}}_t^s$ from learned model $(\hat{g}, \hat{\boldsymbol{f}}^s, \hat{\boldsymbol{f}}^c, \hat{\boldsymbol{p}}^s, \hat{\boldsymbol{p}}^c)$ under Observation Equivalence.

*Proof.* According to Theorem A1, we have

$$[\mathbf{z}_t^s, \mathbf{z}^c] = h([\hat{\mathbf{z}}_t^s, \hat{\mathbf{z}}^c]), \quad (12)$$

where $[.]$ denotes the concatenation operation. The corresponding Jacobian matrix can be formulated as

$$H_t = \begin{bmatrix} \frac{\partial \mathbf{z}_t^s}{\partial \hat{\mathbf{z}}_t^s} & \frac{\partial \mathbf{z}^c}{\partial \hat{\mathbf{z}}_t^s} \\ \frac{\partial \mathbf{z}_t^s}{\partial \hat{\mathbf{z}}^c} & \frac{\partial \mathbf{z}^c}{\partial \hat{\mathbf{z}}^c} \end{bmatrix}. \quad (13)$$

Consider a mapping from $(\mathbf{x}_{t-1}, \hat{\mathbf{z}}_t^s, \hat{\mathbf{z}}^c)$ to $(\mathbf{x}_{t-1}, \mathbf{z}_t^s, \mathbf{z}^c)$ and its Jacobian matrix

$$\begin{bmatrix} \mathbf{I} & \mathbf{0} \\ * & H_t \end{bmatrix}, \quad (14)$$

where $*$ stands for any matrix, and the absolute value of the determination of this Jacobian is $|H_t|$. Therefore $p(\mathbf{x}_{t-1}, \mathbf{z}_t^s, \mathbf{z}^c) = p(\mathbf{x}_{t-1}, \hat{\mathbf{z}}_t^s, \hat{\mathbf{z}}^c)/|H_t|$. Dividing both side by $p(\mathbf{x}_{t-1})$ gives

$$\begin{aligned} & p(\mathbf{z}_t^s, \mathbf{z}^c | \mathbf{x}_{t-1}) = p(\hat{\mathbf{z}}_t^s, \hat{\mathbf{z}}^c | \mathbf{x}_{t-1})/|H_t| \\ \Rightarrow \quad & p(\mathbf{z}_t^s, \mathbf{z}^c | g(\mathbf{z}_{t-1}^s, \mathbf{z}^c)) = p(\hat{\mathbf{z}}_t^s, \hat{\mathbf{z}}^c | \hat{g}(\hat{\mathbf{z}}_{t-1}^s, \hat{\mathbf{z}}^c))/|H_t| \\ \Rightarrow \quad & p(\mathbf{z}_t^s, \mathbf{z}^c | \mathbf{z}_{t-1}^s, \mathbf{z}^c) = p(\hat{\mathbf{z}}_t^s, \hat{\mathbf{z}}^c | \hat{\mathbf{z}}_{t-1}^s, \hat{\mathbf{z}}^c)/|H_t| \\ \Rightarrow \quad & p(\mathbf{z}_t^s | \mathbf{z}_{t-1}^s, \mathbf{z}^c) = p(\hat{\mathbf{z}}_t^s | \hat{\mathbf{z}}_{t-1}^s, \hat{\mathbf{z}}^c)/|H_t| \\ \Rightarrow \quad & p(\mathbf{z}_t^s | \mathbf{z}_{t-1}^s) = p(\hat{\mathbf{z}}_t^s | \hat{\mathbf{z}}_{t-1}^s)/|H_t| \end{aligned} \quad (15)$$

For the first two implications, we utilize the inversion of the mixing function to replace the condition. For the third, since $\mathbf{z}^c$ is conditioned on itself, it remains fixed. For the last one, given that in the generating process, $\epsilon_{t,i}^s$ and $\epsilon_j^c$ for all $i, j, t$ are independently sampled, their disjoint successors are also independent, i.e., $\mathbf{z}_t^s | \mathbf{z}_{t-1}^s \perp \mathbf{z}^c$. Similarly, following assumption C3, we have $\hat{\mathbf{z}}_t^s | \hat{\mathbf{z}}_{t-1}^s \perp \hat{\mathbf{z}}^c$. Thus, we can remove $\mathbf{z}^c$ from the condition.

For simplicity, denote $\eta_t \triangleq \log p(\mathbf{z}_t^s | \mathbf{z}_{t-1}^s)$ and $\eta_{t,i} \triangleq \log p(z_{t,i}^s | \mathbf{z}_{t-1}^s)$ and we have

$$\eta_t = \hat{\eta}_t - \log |H_t|. \quad (16)$$

For any two different $\hat{z}_{t,i}^s, \hat{z}_{t,j}^s \in \mathbf{z}_t^s$, in partial derivative with regard to $\hat{z}_{t,i}^s$ gives

$$\sum_{k=1}^{n_s} \frac{\partial \eta_t}{\partial z_{t,k}^s} \cdot \frac{\partial z_{t,k}^s}{\partial \hat{z}_{t,i}^s} = \frac{\partial \hat{\eta}_t}{\partial \hat{z}_{t,i}^s} - \frac{\partial \log |H_t|}{\partial \hat{z}_{t,i}^s}. \quad (17)$$

Reorganize the left-hand side of Equation 17 with mutual independence of $\mathbf{z}_t^s | \mathbf{z}_{t-1}^s$ yields

$$\sum_{k=1}^{n_s} \frac{\partial \eta_t}{\partial z_{t,k}^s} \cdot \frac{\partial z_{t,k}^s}{\partial \hat{z}_{t,i}^s} = \sum_{k=1}^{n_s} \frac{\partial \prod_{k'=1}^{n_s} \eta_{t,k'}}{\partial z_{t,k}^s} \cdot \frac{\partial z_{t,k}^s}{\partial \hat{z}_{t,i}^s} = \sum_{k=1}^{n_s} \frac{\partial \eta_{t,k}}{\partial z_{t,k}^s} \cdot \frac{\partial z_{t,k}^s}{\partial \hat{z}_{t,i}^s}, \quad (18)$$

and we have

$$\sum_{k=1}^{n_s} \frac{\partial \eta_{t,k}}{\partial z_{t,k}^s} \cdot \frac{\partial z_{t,k}^s}{\partial \hat{z}_{t,i}^s} = \frac{\partial \hat{\eta}_t}{\partial \hat{z}_{t,i}^s} - \frac{\partial \log |H_t|}{\partial \hat{z}_{t,i}^s}. \tag{19}$$

Further get the second-order derivative with regard to $\hat{z}_{t,j}^s$ as

$$\sum_{k=1}^{n_s} \frac{\partial^2 \eta_{t,k}}{\partial^2 z_{t,k}^s} \cdot \frac{\partial z_{t,k}^s}{\partial \hat{z}_{t,i}^s} \cdot \frac{\partial z_{t,k}^s}{\partial \hat{z}_{t,j}^s} + \sum_{k=1}^{n_s} \frac{\partial \eta_{t,k}}{\partial z_{t,k}^s} \cdot \frac{\partial^2 z_{t,k}^s}{\partial \hat{z}_{t,i}^s \partial \hat{z}_{t,j}^s} = \frac{\partial^2 \hat{\eta}_t}{\partial \hat{z}_{t,i}^s \partial \hat{z}_{t,j}^s} - \frac{\partial^2 \log |H_t|}{\partial \hat{z}_{t,i}^s \partial \hat{z}_{t,j}^s}. \tag{20}$$

Next, using the mutual independence of $\hat{\mathbf{z}}_t^s | \hat{\mathbf{z}}_{t-1}^s$ in assumption C3, we have $\frac{\partial^2 \hat{\eta}_t}{\partial \hat{z}_{t,i}^s \partial \hat{z}_{t,j}^s} = 0$ according to the connection between conditional independence and cross derivatives (Lin, 1997). Thus we have

$$\sum_{k=1}^{n_s} \frac{\partial^2 \eta_{t,k}}{\partial^2 z_{t,k}^s} \cdot \frac{\partial z_{t,k}^s}{\partial \hat{z}_{t,i}^s} \cdot \frac{\partial z_{t,k}^s}{\partial \hat{z}_{t,j}^s} + \sum_{k=1}^{n_s} \frac{\partial \eta_{t,k}}{\partial z_{t,k}^s} \cdot \frac{\partial^2 z_{t,k}^s}{\partial \hat{z}_{t,i}^s \partial \hat{z}_{t,j}^s} = - \frac{\partial^2 \log |H_t|}{\partial \hat{z}_{t,i}^s \partial \hat{z}_{t,j}^s}. \tag{21}$$

Now we get the third-order derivative with regard to any $z_{t-1,l}^s$ as

$$\sum_{k=1}^{n_s} \frac{\partial^3 \eta_{t,k}}{\partial^2 z_{t,k}^s \partial z_{t-1,l}^s} \cdot \frac{\partial z_{t,k}^s}{\partial \hat{z}_{t,i}^s} \cdot \frac{\partial z_{t,k}^s}{\partial \hat{z}_{t,j}^s} + \sum_{k=1}^{n_s} \frac{\partial \eta_{t,k}}{\partial z_{t,k}^s} \cdot \frac{\partial^2 z_{t,k}^s}{\partial \hat{z}_{t,i}^s \partial \hat{z}_{t,j}^s} = 0, \tag{22}$$

where we use the property that the entries of $H_t$ do not depend on $z_{t-1,l}^s$.

Given assumption C2, there exists $2n_s$ different values of $z_{t-1,l}^s$ such that the $2n_s$ vectors $\mathbf{v}_{t,l}$ linearly independent. The only solution to Equation 22 is to set

$$\frac{\partial z_{t,k}^s}{\partial \hat{z}_{t,i}^s} \cdot \frac{\partial z_{t,k}^s}{\partial \hat{z}_{t,j}^s} = 0, \frac{\partial^2 z_{t,k}^s}{\partial \hat{z}_{t,i}^s \partial \hat{z}_{t,j}^s} = 0. \tag{23}$$

According to Theorem A1, the blockwise identifiability is established. Thus,

$$H_t = \begin{bmatrix} \frac{\partial \mathbf{z}_t^s}{\partial \hat{\mathbf{z}}_t^s} & \frac{\partial \mathbf{z}_t^c}{\partial \hat{\mathbf{z}}_t^s} \\ \frac{\partial \mathbf{z}_t^s}{\partial \hat{\mathbf{z}}^c} & \frac{\partial \mathbf{z}_t^c}{\partial \hat{\mathbf{z}}^c} \end{bmatrix} \tag{24}$$

is invertible, with $\frac{\partial \mathbf{z}^c}{\partial \hat{\mathbf{z}}_t^s} = 0$ and $\frac{\partial \mathbf{z}_t^s}{\partial \hat{\mathbf{z}}_t^s}$ has at most one nonzero element in each row and each column.

Thus, we have

$$H_t = \begin{bmatrix} \frac{\partial \mathbf{z}_t^s}{\partial \hat{\mathbf{z}}_t^s} & 0 \\ \frac{\partial \mathbf{z}_t^s}{\partial \hat{\mathbf{z}}^c} & \frac{\partial \mathbf{z}^c}{\partial \hat{\mathbf{z}}^c} \end{bmatrix} \tag{25}$$

and $\frac{\partial \mathbf{z}_t^s}{\partial \hat{\mathbf{z}}_t^s}$ must have one and only one non-zero entry in each column and row.

$\square$

## A.2 DISCUSSION OF ASSUMPTIONS

In this section, we give a brief discussion of the assumptions in the real world scenarios.

Assumption B1 requires a smooth distribution, which is usually hold in the real world. Secondly, Assumption B2 imposes a minimal requirement on the number of variables. The linear operator $L_{b|a}$ ensures that there is sufficient variation in the density of $b$ for different values of $a$, thereby guaranteeing injectivity. In a video, $\mathbf{x}_t$ is of much higher dimensionality compared to the latent variables. As a result, the injectivity assumption is easily satisfied. In practice, following the principle of minimal changes, if a model with fewer latent variables can successfully achieve observational equivalence, it is more likely to learn the true distribution. Assumption B3 requires the distribution of $\mathbf{x}_t$ changes when the value of latent variables changes. This assumption is much weaker compared to the widely used invertibility assumption adopted by previous works, such as (Yao et al., 2022). The aforementioned assumptions concern the underlying data-generating process. The aforementioned

assumptions pertain to the underlying data-generating process, and are often easily satisfied in real-world video scenarios. Given a model that can explicitly disentangle motion and identity while maintaining high generative fidelity, block-wise disentanglement becomes a natural and attainable property under these conditions. This also explains why models such as MoCoGAN (Tulyakov et al., 2018) are able to effectively separate motion and identity.

Assumption C1 further requires the underlying distribution to be smooth, while Assumptions C2 and C3 impose mild conditions on the variability of latent factors. These assumptions are not overly restrictive and are often satisfied in practice. Notably, even when the sufficiency conditions are satisfied only by a subset of latent variables, or when mutual independence is partially violated, identifiability can still be achieved at the subspace level, as demonstrated by (Kong et al., 2023) in Section A.1. To further attain disentanglement at the component-wise level, the architectural design of our proposed Temporal Transition Module (TTM) is essential.

### A.3 EXAMPLES OF INJECTIVE LINEAR OPERATORS

The assumption that a linear operator is injective is commonly used in nonparametric identification (Hu & Schennach, 2008b; Carroll et al., 2010; Hu & Shum, 2012). Intuitively, this means that distinct input distributions of a linear operator correspond to distinct output distributions. To clarify this assumption, we provide several examples illustrating the mapping from $p_{\mathbf{a}} \to p_{\mathbf{b}}$, where $\mathbf{a}$ and $\mathbf{b}$ are random variables.

*Example* 1 (**Inverse Transformation**). $b = g(a)$, where $g$ is an invertible function.

*Example* 2 (**Additive Transformation**). $b = a + \epsilon$, where the distribution $p(\epsilon)$ must not vanish entirely under the Fourier transform (Theorem 2.1 in (Mattner, 1993)).

*Example* 3. $b = g(a) + \epsilon$, requiring the same conditions as in Examples 1 and 2.

*Example* 4 (**Post-linear Transformation**). $b = g_1(g_2(a) + \epsilon)$, a post-nonlinear model with invertible nonlinear functions $g_1$ and $g_2$, combining the assumptions in Examples 1–3.

*Example* 5 (**Nonlinear Transformation with Exponential Family**). $b = g(a, \epsilon)$, where the joint distribution $p(a, b)$ belongs to an exponential family.

*Example* 6 (**General Nonlinear Transformation**). $b = g(a, \epsilon)$, representing a general nonlinear mapping. Certain deviations from the nonlinear additive model (Example 3), such as polynomial perturbations, can still be tractable.

### A.4 EXTENSION TO NONSTATIONARY SCENARIOS

When the generative process is nonstationary, meaning that all latent variables are conditioned on an exogenous variable $u$, we obtain the nonstationary form of Equation 1:

$$z_{t,i}^s = f_i^s\big(\mathbf{Pa}(z_{t,i}^s), \epsilon_{t,i}^s, u\big), \quad z_j^c = f_j^c\big(\epsilon_j^c, u\big), \tag{26}$$

where $u$ may represent weak supervision or textual guidance. As stated in Theorem 2 of (Yao et al., 2022), the identifiability results are easier to establish because variations in $u$ help to better satisfy Assumption C2. More precisely, Equation 11 can be rewritten as

$$\begin{aligned}
\mathbf{v}_{t,l} \triangleq & \Big(\frac{\partial^2(\eta_{t,1}(u) - \eta_{t,1}(u'))}{\partial z_{t,1} \partial z_{t-1,l}}, \cdots, \frac{\partial^2(\eta_{t,n_s}(u) - \eta_{t,n_s}(u'))}{\partial z_{t,n_s} \partial z_{t-1,l}}\Big) \\
& \oplus \Big(\frac{\partial^3(\eta_{t,1}(u) - \eta_{t,1}(u'))}{\partial^2 z_{t,1} \partial z_{t-1,l}}, \cdots, \frac{\partial^3(\eta_{t,n_s}(u) - \eta_{t,n_s}(u'))}{\partial^2 z_{t,n_s} \partial z_{t-1,l}}\Big),
\end{aligned} \tag{27}$$

where $\eta_{t,i}(u) \triangleq \log p(z_{t,i}^s \mid \mathbf{z}_{t-1}^s, u)$, and $u$ and $u'$ denote two different exogenous conditions.

### A.5 EXTENSION TO CONTENT AND STYLE ENTANGLEMENT

For example, the motion patterns of a fish differ from those of a human, suggesting that $z^c$ could affect $z^s$. In such cases, $z^s$ can still be reformulated as a function of $z^c$ and an independent noise term (Zhang et al., 2015). Consequently, the noise term can be regarded as the new dynamic variable $z^s$. Therefore, the video can still be interpreted as being generated from a fixed $z^c$ and a varying, independent $z^s$.

## B    RELATED WORKS

### B.1    CONTROLLABLE VIDEO GENERATION

Recent advances in controllable video generation have led to significant progress, with text-to-video (T2V) models (Yang et al., 2024b; Singer et al., 2022; Ho et al., 2022; Zhou et al., 2022; Zheng et al., 2024) achieving impressive results in generating videos from textual descriptions. However, effectiveness of the control is highly dependent on the quality of the input prompt, making it difficult to achieve fine-grained control over the generated content. An alternative method for control involves leveraging side information such as pose (Tu et al., 2024; Zhu et al., 2025), camera motion (Yang et al., 2024a; Wang et al., 2024), depth (Liu et al., 2024; Xing et al., 2024) and so on. While this approach allows for more precise control, it requires paired data, which can be challenging to collect. Besides, most of the aforementioned alignment-based techniques share a common issue: the control signals are directly aligned with the entire video. This issue not only reduces efficiency but also complicates the task of achieving independent control over different aspects of the video, which further motivates us to propose a framework to find the disentanglement representation for conditional generation.

### B.2    NONLINEAR INDEPENDENT COMPONENT ANALYSIS

Nonlinear independent component analysis offers a potential approach to uncover latent causal variables in time series data. These methods typically utilize auxiliary information, such as class labels or domain-specific indices, and impose independence constraints to enhance the identifiability of latent variables. Time-contrastive learning (TCL) (Hyvarinen & Morioka, 2016) builds on the assumption of independent sources and takes advantage of the variability in variance across different segments of data. Similar Permutation-based contrastive learning (PCL) (Hyvarinen & Morioka, 2017) introduces a learning framework that tell true independent sources from their permuted counterparts. Additionally, i-VAE (Khemakhem et al., 2020) employs deep neural networks and Variational Autoencoders (VAEs) to closely approximate the joint distribution of observed data and auxiliary non-stationary regimes. Recently, (Yao et al., 2021; 2022) extends the identifiability to linear and nonlinear non-Gaussian cases without auxiliary variables, respectively. CaRiNG (Chen et al., 2024) further tackles the case when the mixing process is non-invertible. Additionally, CITRIS (Lippe et al., 2022; 2023) emphasizes the use of intervention target data, and IDOL (Li et al., 2024c) incorporates sparsity into the latent transition process to identify latent variables, even in the presence of instantaneous effects.

## C    DATASET DETAILS

We use the following datasets to verify our model.

- FaceForensics (Rössler et al., 2018): A forensics dataset consisting of 1,000 original video sequences. All videos contain a trackable, mostly frontal face without occlusions.
- SkyTimelapse (Xiong et al., 2018): Typically consists of sequential images or videos capturing the dynamic behavior of the sky over time. We use the 2,368 officially released videos.
- RealEstate10K (Zhou et al., 2018): A large dataset of camera poses corresponding to 10 million frames from about 80,000 video clips, gathered from around 10,000 YouTube videos. Each clip's poses form a trajectory specifying camera position and orientation, derived via SLAM and bundle adjustment. To our knowledge, our method is the first GAN-based approach to leverage this dataset for unconditional video generation.
- CelebV-HQ (Zhu et al., 2022): A large-scale, high-quality video dataset with diverse celebrity identities and actions. It contains 35,666 video clips with a minimum resolution of 512×512, covering 15,653 identities.

## D    METRIC DETAILS

We use the following metrics to verify our model.

- FVD (Fréchet Video Distance (Unterthiner et al., 2019)) is a metric for evaluating video generation quality. It compares real and generated videos by extracting spatiotemporal features with a pretrained model and computing the Fréchet distance between their feature distributions.

- SAP (Separated Attribute Predictability (Kumar et al., 2017)) measures how well each latent dimension is associated with a single ground-truth factor. It's computed by training a simple regressor to predict true factors from each latent dimension, then comparing the two best-performing dimensions. A higher score means better disentanglement.

- Modularity (Ridgeway & Mozer, 2018) measures whether each latent dimension encodes information about at most one true factor. It penalizes when a single latent variable carries mixed information about multiple true factors. Perfect modularity means each latent dimension corresponds only to a single factor.

- MCC (Mean Correlation Coefficient) is a commonly used disentanglement metric in representation learning. Let $Z \in \mathbb{R}^D$ be the ground-truth latent vector and $\widehat{Z} \in \mathbb{R}^{\widehat{D}}$ the estimated vector. To calculate MCC, we first compute the Pearson correlations $R_{ij} = \mathrm{corr}(Z_i, \widehat{Z}_j)$, then select an injective matching $\pi : \{1, \ldots, D\} \to \{1, \ldots, \widehat{D}\}$ maximizing $\sum_{i=1}^{D} |R_{i,\pi(i)}|$. Finally, the MCC value is defined as $\mathrm{MCC} = \frac{1}{D} \sum_{i=1}^{D} |R_{i,\pi(i)}|$.

## E  REPRODUCTABILITY

All of our models are trained on an NVIDIA A100 40G GPU. Our model requires around a throughput of 20 million frames for convergence. For the baseline models, we use the official implementation with the default hyperparameters. The configuration of hyperparameters for our model training is as shown in Table 4.

Table 4: Configuration details for the model and training setup.

| Name | Variable Name | Value/Description |
|---|---|---|
| Information regularization term weight | lambda_KL | 1 |
| Dimensionality of $\mathbf{z}^c$ | - | 512 |
| Dimensionality of GRU | - | 64 |
| Dimensionality of $\mathbf{z}_s^t$ | - | FaceForensics: 4, SkyTimelapse: 12, RealEstate: 12 |
| Batch size (per GPU) | batch | 16 |
| Conditional mode | cond_mode | flow |
| Flow normalization | flow_norm | 1 |
| Sparsity weight | lambda_sparse | 0.1 |
| Number of input channels | channel | 3 |
| Number of mapping layers | num_layers | 8 |
| Label embedding features | embed_features | 512 |
| Intermediate layer features | layer_features | 512 |
| Activation function | activation | lrelu |
| Learning rate multiplier | lr_multiplier | 0.01 |
| Moving average decay | w_avg_beta | 0.995 |
| Discriminator architecture | - | resnet |
| Channel base | channel_base | 32768 |
| Maximum number of channels | channel_max | 512 |

The architecture of our proposed Temporal Transition Module is shown in Table 5.

## F  MORE EXPERIMENTS

### F.1  LONGER VIDEOS

We also verify our methods on generating longer videos with 32 frames on the FaceForensics Dataset, as shown in Table 6. Our method achieves substantially better performance than MoStGAN-V on disentanglement metrics.

Table 5: Architecture of Temporal Transition Module.

| Component | Structure |
|---|---|
| MappingNetwork.gru | $4 \times 256 \times 3$ |
| MappingNetwork.h_to_c | FullyConnectedLayer $(256 \times 64)$ |
| MappingNetwork.embed | FullyConnectedLayer $(64 \times 512)$ |
| MappingNetwork.flow.model.0 | DenseSigmoidFlow |
| MappingNetwork.flow.model.1 | DenseSigmoidFlow |
| MappingNetwork.flow_fc0 | FullyConnectedLayer $(512 \times 512)$ |
| MappingNetwork.flow_fc1 | FullyConnectedLayer $(512 \times 512)$ |
| MappingNetwork.flow_fc2 | FullyConnectedLayer $(512 \times 284)$ |
| FullyConnectedLayer | FullyConnectedLayer $(512 \times 512)$ |
| FullyConnectedLayer | FullyConnectedLayer $(512 \times 512)$ |

Table 6: Performance on FaceForensics with length=32.

| Metrics | MoStGAN-V | CoVoGAN |
|---|---|---|
| FVD $\downarrow$ | 159.24 | **145.77** |
| MCC (%) $\uparrow$ | 14.23 | **29.99** |
| SAP (%) $\uparrow$ | 0.87 | **4.20** |
| Modularity (%) $\uparrow$ | 8.15 | **10.14** |

## F.2 CHOICE OF $n_s$

In CoVoGAN, the dimension of $n_s$ is treated as a tunable hyperparameter. While the choice of $n_s$ is important, it turns out to be not overly sensitive. From a theoretical standpoint, assuming a known latent dimensionality is standard in the nonlinear ICA literature (Hyvarinen & Morioka, 2016; Kong et al., 2023). Moreover, even when $n_s$ is not specified exactly, the model can still yield meaningful results: if $n_s$ is set too large, the model tends to fit some noise, whereas if it is set too small, certain components may be merged. This explains why our empirical results show robustness with respect to $n_s$. As presented in Table 7, varying it within a reasonable range does not lead to significant performance changes.

Table 7: Ablation on $n_s$ on the FaceForensics dataset.

| $n_s$ | 4 | 8 |
|---|---|---|
| FVD $\downarrow$ | 48.8 | **47.9** |

## F.3 SUPERVISED VIDEO GENERATION

In this paper, we propose a method for controlling video generation without relying on specific supervisory signals such as text or trajectories. Our goal is not to surpass supervised approaches like text-to-video or trajectory-guided models, but rather to explore the feasibility of achieving disentangled video generation in the absence of explicit supervision. This setting is considerably more challenging, as annotations are typically task-specific and costly to obtain. Moreover, when appropriate supervisory signals such as text or trajectories are available, our method can be combined with them to enhance controllability through an identifiable generative process.

To further strengthen our evaluation, we have integrated our TTM module into two additional models: the diffusion-based Text-to-Video generation model, Wan (Wan et al., 2024), and the trajectory-supervised model, MotionCtrl (Wang et al., 2024).

For text-to-video generation, we integrate our TTM module into the Wan-1.3B model and finetune it on a subset of WebVid-10M (Bain et al., 2021), using the first 10% of the data. We conduct a thorough comparison against the original Wan-1.3B model and CogVideoX-2B (Yang et al., 2024b). The results, presented in Table 8, were evaluated using the VBench (Huang et al., 2023) toolkit for the first six metrics, and three different implementations of CLIP similarity: hf_clip_vit_b32,

hf_clip_vit_l14 (Radford et al., 2021), hf_laion_clip_vit_b32 (Rössler et al., 2018) for the controllability measures. While our model shows a slight decrease in some metrics compared to CogVideoX-2B (e.g., hf_clip_vit_b32), it still outperforms the original Wan-1.3B model, especially to the aspect of controllability. Examples of generated videos are provided in Figure 9.

Table 8: Comparison of text-to-video generation across multiple metrics (larger is better).

| Metric | CogVideoX-2B | Wan-1.3B | TTM + Wan-1.3B |
|---|---|---|---|
| imaging_quality ↑ | 0.4870 | 0.6851 | **0.6877** |
| motion_smoothness ↑ | 0.9886 | 0.9874 | **0.9935** |
| dynamic_degree ↑ | 0.3465 | 0.3960 | **0.4059** |
| subject_consistency ↑ | 0.9347 | 0.9324 | **0.9529** |
| background_consistency ↑ | 0.9585 | 0.9375 | **0.9628** |
| aesthetic_quality ↑ | 0.3849 | **0.5370** | 0.4924 |
| hf_clip_vit_b32 ↑ | **0.2428** | 0.2234 | 0.2288 |
| hf_clip_vit_l14 ↑ | **0.2007** | 0.1788 | 0.1892 |
| hf_laion_clip_vit_b32 ↑ | 0.1859 | 0.1791 | **0.1980** |

We also integrate our TTM module into the MotionCtrl model and conducted a comparison on the RealEstate10K dataset, with regard to video generation conditioned on trajectory of camera poses. As shown in Table 9, our TTM module successfully improves the performance of the MotionCtrl model.

Table 9: Comparison of MotionCtrl and TTM + MotionCtrl across multiple metrics (lower is better).

| Metric | MotionCtrl | TTM + MotionCtrl |
|---|---|---|
| CamMC ↓ | 0.0840 | **0.0776** |
| FID ↓ | 130.29 | **129.05** |
| FVD ↓ | 934.37 | **917.28** |

Since both baseline models are diffusion-based rather than GAN-based, we integrate the TTM module through cross-attention as a conditioning mechanism, following a similar strategy to that used in an identifiable image generation model (Xie et al., 2025). Although this design is not fully aligned with our theoretical framework, it highlights a promising direction for future investigation.

## F.4   COMPUTATIONAL EFFICIENCY

We also compare the computational efficiency with that of the baselines. The size of our model's generator is comparable to that of StyleGAN2-ADA but significantly smaller than the generators of the baselines. Additionally, the inference time of our model is much faster, as shown in Table 10.

## F.5   MORE VISUALIZATION RESULTS

In this section, we provide more qualitative video results generated by our approach. As shown in Figure 6, we compare the generative quality of the videos among all models. As can be seen in Figure 7, our method can control different identities of sky scenes with consistent constructed motions.

## F.6   EXPERIMENTS ON HUMAN MOTION DATASET

We also conduct experiments on more TaiChi (Sun et al., 2017), a more complex dataset with richer human motion dynamics. We compare FVD, MCC, SAP, and Modularity on the TaiChi $64^2$ dataset. For the disentanglement metrics, we use the open-source package MediaPipe to extract human motion keypoints as ground-truth variables. However, for MoStGAN-V, MediaPipe fails to extract any keypoints from the generated videos due to poor generation quality, making the disentanglement metrics inapplicable for this method. The quality results and quantity results can be found in Figure 8 and Table 11 respectively.

Table 10: Comparison of different models in terms of generator parameters (Params, in millions) and the time (Time, in seconds) required to generate 16-frame 2048 videos with a resolution of $256 \times 256$.

| Method | Params (M) | Time (s) |
|---|---|---|
| StyleGAN2-ADA | 23.19 | - |
| DIGAN | 69.97 | 142.06 |
| StyleGAN-V | 32.11 | 181.45 |
| MoStGAN-V | 40.92 | 207.11 |
| Latte | 757.38 | 45340.74 |
| CoVoGAN (ours) | **24.98** | **98.93** |

Table 11: Comparison of StyleGAN-V, MoStGAN-V, and CoVoGAN on Taichi.

| Metrics on Taichi | StyleGAN-V | MoStGAN-V | CoVoGAN |
|---|---|---|---|
| FVD $\downarrow$ | 68.52 | 60.39 | **55.3** |
| MCC (%) $\uparrow$ | 18.2 | - | **42.3** |
| SAP (%) $\uparrow$ | 1.5 | - | **3.2** |
| Modularity (%) $\uparrow$ | 0.2 | - | **1.9** |

## F.7 EXPERIMENTS ON INDEPENDENT UTILITY OF $z^c$ AND $z^s$

In this subsection, we verify that the $z^c$ and $z^s$ can be utilized independently to generate videos. We have added the following ablation experiments: we first randomly sample $z^c$ and $z^s$ independently from the learned distribution, then for each $z^c$, we use different $z^s$ to generate videos, and vice versa. We compare the FVD and disentanglement metrics and both show that the two representations can be used independently to generate videos with good performance. The results are shown in Table 12.

## F.8 EXPERIMENTS WITH MULTI-DOMAINS

To further verify our model on a more complex dataset with multiple domains, we conduct experiments on UCF-101 (Soomro et al., 2012), which contains realistic videos of 101 human action classes. The dataset includes over 13k clips and a total of 27 hours of video data. We evaluate our model with an increasing number of action categories to demonstrate its ability to generalize across multiple-domain scenarios.

We train and evaluate both our model and the baseline as conditional generation models, following the generative process described in Equation 26. Experiments are conducted on three subsets of the UCF-101 dataset: 3 classes (ApplyLipstick, BodyWeightSquats, PlayingGuitar), 20 classes (BabyCrawling, BalanceBeam, Billiards, BlowingCandles, BodyWeightSquats, BoxingSpeedBag, Fencing, FieldHockeyPenalty, FloorGymnastics, FrontCrawl, Hammering, MoppingFloor, PlayingSitar, PullUps, PushUps, Rowing, ShavingBeard, Skijet, SumoWrestling, Typing), and all 101 classes. We downsample the resolution to $64^2$ due to GPU shortage. Since it is impractical to use keypoints as ground truth for controllability evaluation when multiple actions are involved, we instead assess controllability by measuring the classification accuracy of generated videos with respect to their target action classes. The classifier is implemented as a simple CNN. The performance results are summarized in Table 13. The baseline model crashed during training when class number is larger than 3, so we list the results on 3 classes only. Our model outperforms the baseline model for both fidelity and controllability. Randomly sampled videos trained on 3 classes setting are shown in Figure 10.

## F.9 EXPERIMENTS ON BLOCKWISE IDENTIFICATION

To evaluate the blockwise identification, we further assess the consistency of motion when only the content latent code is modified. Experiments are conducted on the FaceForensics dataset ($256^2$). Specifically, we randomly sample a video and then alter its $z^c$ to generate another video. We extract facial keypoints from both videos using dlib and compute the average keypoint distance (AKD) between them. A lower AKD indicates stronger disentanglement between content and motion. The results are summarized in Table 14.

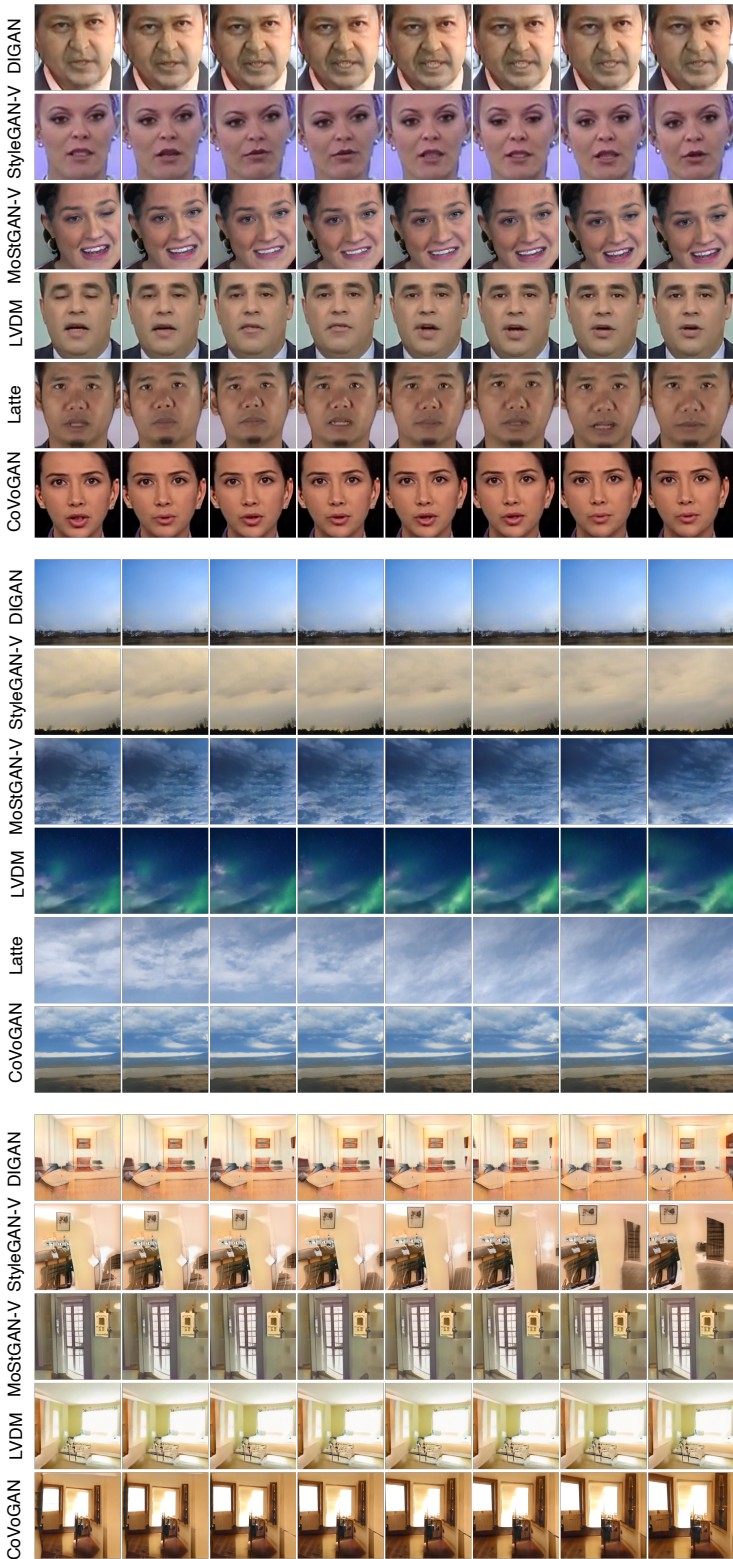

Figure 6: Random samples from the comparison baselines and our model on real-world datasets FaceForensics $256^2$, SkyTimelapse $256^2$, RealEstate $256^2$. Start from t = 0 and report every second frame from a 16-frame video clip.

Table 12: Extrapolation ablation study on FaceForensics.

| Metrics | Vanilla CoVoGAN | Varying $z^s$ | Varying $z^c$ |
|---|---|---|---|
| MCC (%) ↑ | 33.78 | **36.89** | 31.76 |
| SAP (%) ↑ | 8.48 | 8.71 | **10.39** |
| Modularity (%) ↑ | **17.37** | 14.75 | 11.29 |

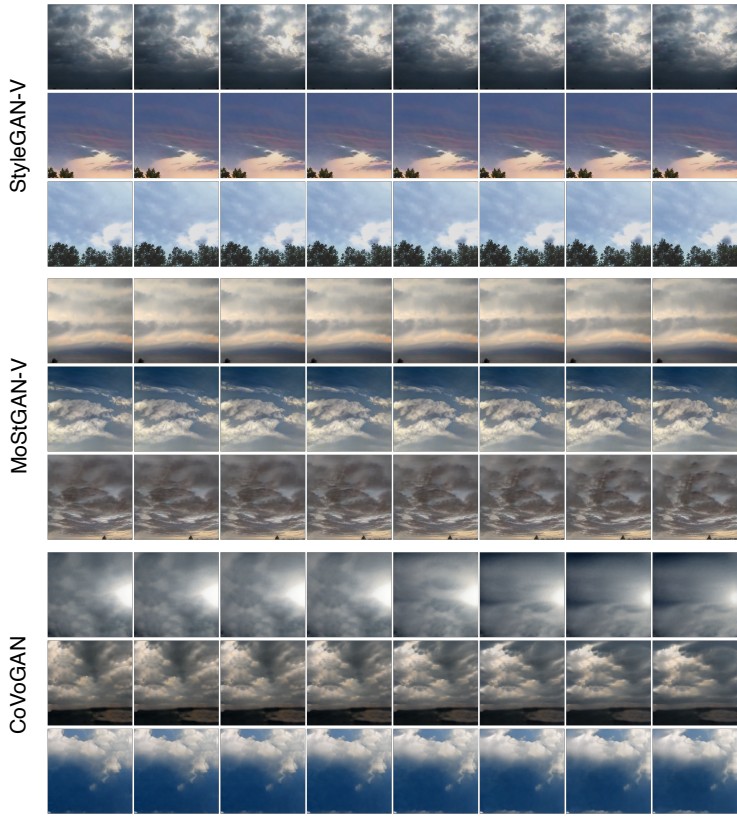

Figure 7: Generate same motion of different identities on SkyTimelapse Dataset.

## G  GANS VERSUS DIFFUSION MODELS

In this section, we explain the reason that we adopt the GAN framework as the primary implementation in our work. The key difference between our GAN-based framework and diffusion-based approaches lies mainly in the training objective rather than the architecture. To illustrate, consider an ideal scenario where both videos and their true latent factors are observable. In this fully supervised case, the model could be trained in a straightforward, sample-to-sample manner. However, under the unsupervised setting considered in this work, the latent variables are unobserved, and the model can

| Class No. | Metric | FVD ↓ | Accuracy ↑ |
|---|---|---|---|
| 3 | MoStGAN-V | 897.86 | 0.35 |
| | CoVoGAN | 178.03 | 0.64 |
| 20 | CoVoGAN | 790.23 | 0.10 |
| 101 | CoVoGAN | 1050.61 | - |

Table 13: Experiments on UCF-101 with different numbers of action classes. FVD evaluates video quality, and classification accuracy measures controllability of the generated videos.

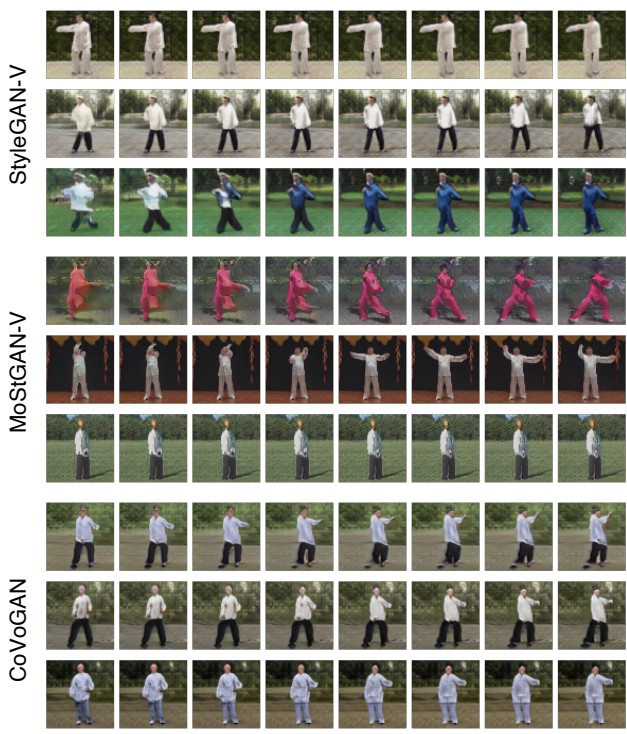

Figure 8: Generate same motion of different identities on Taichi Dataset.

**Text Prompt**: A male dentist explains the surgery using a plaster model of the mandible, holding a dental mock-up and tools while speaking with an elderly patient.

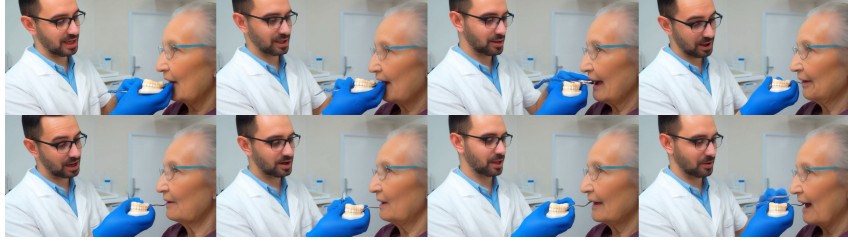

**Text Prompt**: A cheetah rests on the ground, licking its lips while scanning its surroundings

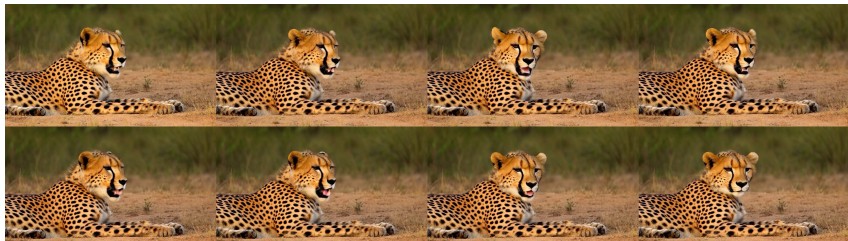

Figure 9: Controllable text-to-video generation results.

learn the mapping by instead ensuring observational equivalence, i.e., aligning the distributions of generated and real videos. This naturally motivates the use of a GAN framework, which enforces such distributional alignment.

Moreover, our theoretical guarantees established in our framework are also compatible with diffusion-based generative processes. In practice, one potential solution is to adopt a GAN-style loss for matching the marginal video distributions while retaining a diffusion-based architecture. Alternatively,

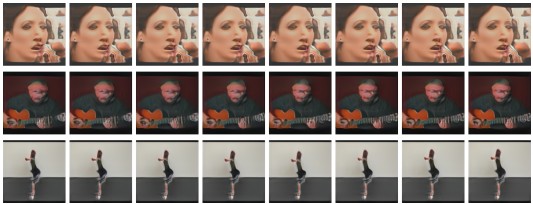

Figure 10: Random samples of generated videos with class: ['ApplyLipstick', 'PlayingGuitar', 'BodyWeightSquats'], from top to bottom.

| Method | AKD ↓ |
|---|---|
| StyleGAN-V | 19.53 |
| MoStGAN-V | 15.86 |
| **CoVoGAN (Ours)** | **6.45** |

Table 14: Comparison of AKD on the FaceForensics dataset. Lower means better content-motion disentanglement.

it is possible to predict the independent noise component directly from video sequences, enabling sample-to-sample training that preserves extra supervision signals, as demonstrated in (Xie et al., 2025). We implement the latter one in our experiments in Appendix F.3.

# H  IMPACT STATEMENTS

This study introduces both a theoretical framework and a practical approach for extracting disentangled causal representations from videos. Such advancements enable the development of more transparent and interpretative models, enhancing our grasp of causal dynamics in real-world settings. This approach may benefit many real-world applications, including healthcare, auto-driving, content generation, marketing and so on.

