# OpenReview forum: "Controllable Video Generation with Provable Disentanglement"
_ICLR.cc/2026/Conference — ICLR 2026 Poster_

### Official Review · Reviewer_7TFq · 2025-11-01

**Soundness:** 3
**Presentation:** 3
**Contribution:** 3
**Rating:** 6
**Confidence:** 2

**Summary:**

The paper introduces CoVoGAN, a GAN-based framework for controllable video generation with provable disentanglement between content (static) and motion (dynamic) factors. the authors propose to add a light Temporal Transition Module (GRU + component-wise flow) in front of a StyleGAN-like generator, plus a video discriminator and an MI term to tie dynamics to what is being produced.

**Strengths:**

- The authors provided solid theory to back their claims
- The paper is well-written and easy to follow

**Weaknesses:**

- The authors provide a disentanglement comparisons between all method, yet it is likely favor methods with compact latents (i.e Diffusion models vs GANs)
- while the author demonstrate independence between content and motion, I could not find any table quantifying this improvement/
- The method is based of GANs. However, due the popularity of diffusion models recently, the contribution on GANs might be undermind.

**Questions:**

See above.

---

> ### Author Response · Authors · 2025-11-23
>
> Dear Reviewer 7TFq, thank you for your insightful suggestions, which helped us better demonstrate the blockwise identification amd improve the clarity of our contributions.
>
> >**W1:** The authors provide a disentanglement comparisons between all method, yet it is likely favor methods with compact latents (i.e Diffusion models vs GANs).
>
> **A1:** Thank you for your insightful comment. Indeed, the lack of a compact latent representation in a vanilla diffusion model like LVDM is indeed one of the key motivations behind our proposed approach, since one cannot easily control the video by manipulating a component of the latent code in this case. We compare it here to highlight this point. If you think it may cause confusion, we are happy to remove this comparison or move it to the appendix.
>
> >**W2:** While the author demonstrate independence between content and motion, I could not find any table quantifying this improvement.
>
> **A2:** We appreciate your valuable suggestion. In light of you suggestion, we further assess the consistency of motion when only the content latent code is modified. Experiments are conducted on the FaceForensics dataset ($256^2$). Specifically, we randomly sample a video and then alter its $z^c$ to generate another video. We extract facial keypoints from both videos using dlib and compute the average keypoint distance (AKD) between them. A lower AKD indicates stronger disentanglement between content and motion. Here we attach the results for your reference.
>
> | **Method**      | **AKD ↓**  |
> | --------------- | ---------- |
> | StyleGAN-V      | 19.53      |
> | MoStGAN-V       | 15.86      |
> | CoVoGAN (Ours)  | **6.45**   |
>
> We have added this experiment to Appendix F.9 (line 1289) in the revised to paper for better completeness.
>
> >**W3:** The method is based of GANs. However, due the popularity of diffusion models recently, the contribution on GANs might be undermind.
>
> **A3:** Thank you for the insightful suggestion. Infact, our theoretical guarantees established in our framework are general and are also compatible with diffusion-based generative processes. We have already included experiments integrating our proposed method with diffusion-based models in the original submission (see Appendix F.3 on line 1172 in the revised paper). The results demonstrate that our approach can be effectively combined with diffusion models, achieving improved controllability while maintaining high visual fidelity. Here we attach the results for your reference.
>
> | **Metric**               | **CogVideoX-2B** | **Wan-1.3B** | **TTM + Wan-1.3B** |
> | ------------------------ | ---------------- | ------------ | ------------------ |
> | imaging_quality ↑        | 0.4870           | 0.6851       | **0.6877**         |
> | motion_smoothness ↑      | 0.9886           | 0.9874       | **0.9935**         |
> | dynamic_degree ↑         | 0.3465           | 0.3960       | **0.4059**         |
> | subject_consistency ↑    | 0.9347           | 0.9324       | **0.9529**         |
> | background_consistency ↑ | 0.9585           | 0.9375       | **0.9628**         |
> | aesthetic_quality ↑      | 0.3849           | **0.5370**   | 0.4924             |
> | hf_clip_vit_b32 ↑        | **0.2428**       | 0.2234       | 0.2288             |
> | hf_clip_vit_l14 ↑        | **0.2007**       | 0.1788       | 0.1892             |
> | hf_laion_clip_vit_b32 ↑  | 0.1859           | 0.1791       | **0.1980**         |
>
> We now explain why we adopt the GAN framework as the primary implementation in our work. The key difference between our GAN-based framework and diffusion-based approaches lies mainly in the training objective rather than the architecture. To illustrate, consider an ideal scenario where both videos and their true latent factors are observable. In this fully supervised case, the model could be trained in a straightforward, sample-to-sample manner. However, under the unsupervised setting considered in this work, the latent variables are unobserved, and the model must instead ensure observational equivalence, i.e., aligning the distributions of generated and real videos. This naturally motivates the use of a GAN framework, which enforces such distributional alignment.
>
> We have added this discussion to Appendix G (line 1388) in the revised paper for completeness.
>
> [1] Learning Vision and Language Concepts for Controllable Image Generation.

---

### Official Review · Reviewer_KQV9 · 2025-11-02

**Soundness:** 2
**Presentation:** 3
**Contribution:** 3
**Rating:** 4
**Confidence:** 2

**Summary:**

This paper introduces CoVoGAN, a video generation model that achieves fine-grained control through theoretically-grounded disentanglement. The approach separates videos into static content (identity) and dynamic factors (motion), with component-wise separation of individual motion concepts (e.g., head movement vs. eye blinking). Experiments across multiple datasets show superior video quality and controllability compared to baselines, enabling precise independent manipulation of different video aspects.

**Strengths:**

1. This paper provides strong theoretical foundation. It provide rigorous identifiability theorems for video generation, offering mathematical guarantees for disentanglement at both block-wise and component-wise levels.

2. This paper enables independent manipulation of specific motion concepts (e.g., eye blinking, head movement) without affecting content/identity, demonstrated across multiple datasets.

3. The evaluation is thorough across different metrics and datasets.

**Weaknesses:**

1. The paper assumes a strong assumption: videos decompose cleanly into content (which is static) and dynamic motion. However, many real-world vidoes don't fit this dichotomy. For example, deformable objects have constant but moving parts. The identity and motion are entangled. Lighting changes, and shadow moving cannot fit into these two categories either. I am concerned about the limited generalizability of the proposed method to broader video domains.

2. Performance on complex human motion (TaiChi dataset) shows room for improvement, suggesting challenges with highly dynamic scenes.

**Questions:**

None.

---

> ### Author Response · Authors · 2025-11-23
>
> Dear Reviewer KVQ9, we appreicaite your valuable suggestions, which helped us to better clarify the assumptions and improve the completeness of our experiments.
>
> >**W1:** The paper assumes a strong assumption: videos decompose cleanly into content (which is static) and dynamic motion. However, many real-world vidoes don't fit this dichotomy. For example, deformable objects have constant but moving parts. The identity and motion are entangled. Lighting changes, and shadow moving cannot fit into these two categories either. I am concerned about the limited generalizability of the proposed method to broader video domains.
>
> **A1:** Thank you for your insightful comment. We respectfully argue that this assumption is not as strong as it may seem. For example, in the deformable object scenario you mentioned, the static content can still capture the overall shape and appearance of the object, while the dynamic motion can model the deformations over time. Specifically, we may consider $z^c$ as the $3D$ representation of the object, which remains constant no matter how the object deforms. Similarly, lighting changes and moving shadows can be represented as part of the dynamic motion component, while the underlying scene structure remains in the static content.
>
> For example, the motion patterns of a fish differ from those of a human, suggesting that $z^c$ could affect $z^s$. In such cases, $z^s$ can still be reformulated as a function of $z^c$ and an independent noise term [1]. Consequently, the noise term can be regarded as the new dynamic variable $z^s$. Therefore, the video can still be interpreted as being generated from a fixed $z^c$ and a varying, independent $z^s$.
>
> We have added this discussion to Appendix A.5 (line1020) in the revised manuscript for completeness.
>
> [1] On Estimation of Functional Causal Models: General Results and Application to the Post-Nonlinear Causal Model.
>
>
> >**W2:** Performance on complex human motion (TaiChi dataset) shows room for improvement, suggesting challenges with highly dynamic scenes.
>
> **A2:** We appreciate the reviewer’s valuable observation. We would like to clarify that on more complex datasets such as TaiChi, our method achieves a larger relative performance gain compared to less dynamic datasets. The table below reports the performance improvement ratios of CoVoGAN over StyleGAN-V across different datasets. The results indicate that our method is particularly effective at modeling highly dynamic scenes.
>
> | **Dataset**      | **FVD Ratio $\downarrow$ (%)** | **MCC Ratio $\uparrow$ (%)** | **SAP Ratio $\uparrow$ (%)** | **Modularity Ratio $\uparrow$ (%)** |
> | ---------------- | ---------------- | ---------------- | ---------------- | ----------------------- |
> | FaceForensics    | 92.59            | 116.48            | 199.52            | 226.76                  |
> | TaiChi          | 80.70             | 232.41          | 213.33          | 950.00                   |
>
> For a fair comparison, all models were trained at a resolution of 64×64, which makes the samples in Figure 8 appear less clear. (At 256×256 resolution, most videos generated by the baseline models failed to produce detectable keypoints, making it infeasible to evaluate controllability.) To further demonstrate our model’s capability, we also trained on the TaiChi dataset at 256×256 resolution, the results shows a consistent performance gain.
>
> Here we attched the FVDs on 256x256 TaiChi dataset for your reference.
>
> | **Method**      | **FVD ↓**  |
> | --------------- | ---------- |
> | StyleGAN-V      | 212.37   |
> | MoStGAN-V       | 190.95   |
> | CoVoGAN (Ours)  | **143.41** |

---

### Official Review · Reviewer_S4AE · 2025-11-08

**Soundness:** 2
**Presentation:** 3
**Contribution:** 2
**Rating:** 4
**Confidence:** 5

**Summary:**

The authors propose Controllable Video Generative Adversarial Networks (CoVoGAN), an approach designed to disentangle video concepts to facilitate efficient and independent control over individual factors. CoVoGAN is built upon rigorous theoretical insights derived from nonlinear Independent Component Analysis (ICA), establishing conditions for identifiability. It used to achieve component-wise identifiability of the dynamic latent variables, allowing for disentangled control within the motion itself. It proposed the Temporal Transition Module (TTM), integrated as a plug-in into a GAN architecture (based on StyleGAN2-ADA). Extensive experiments across various datasets (including FaceForensics, SkyTimelapse, RealEstate, and CelebV-HQ) demonstrate that CoVoGAN improves both generative quality (measured by FVD) and controllability (measured by SAP, modularity, and MCC) compared to existing video generation models.

**Strengths:**

1. The model successfully achieves both block-wise identifiability (separating content from motion) and component-wise identifiability (separating distinct factors within motion).
2. CoVoGAN achieves the best performance across disentanglement metrics (MCC, SAP, Modularity) compared to strong baselines like StyleGAN-V, MoStGAN-V, LVDM, and Latte.
3. CoVoGAN is computationally efficient, having a generator parameter count (24.98M) comparable to StyleGAN2-ADA and significantly smaller than other baselines.

**Weaknesses:**

1. The primary implementation and theoretical guarantees are grounded in the GAN framework, how to integrate the idea into current diffusion-based models? Compared to diffusion models, GAN-based approaches still have large visual fidelity gap.
2. Experiments only focus on domain-specific datasets, how will be the performance of CoVoGAN on UCF101 or even larger dataset such as Kinetics? It would be better if the author could show the abilities of the model on more complex datasets.
3. Can CoVoGAN be used in long-term video generation?

**Questions:**

see weaknesses

---

> ### Author Response · Authors · 2025-11-23
> **Part I**
>
> Dear Reviewer S4AE, thank you for your valuable suggestions, which helped us to improve the completeness of experiments and the readness of our paper.
>
> >**W1:** The primary implementation and theoretical guarantees are grounded in the GAN framework, how to integrate the idea into current diffusion-based models? Compared to diffusion models, GAN-based approaches still have large visual fidelity gap.
>
> **A1:** We appreciate the reviewer’s insightful comment. We kindly point out that we have already included experiments integrating our proposed method with diffusion-based architectures in the original submission (Appendix F.3, line 1172 in the revised version). The results demonstrate that our approach can be effectively combined with diffusion models, achieving improved controllability while maintaining high visual fidelity. Here we attach the results for your reference.
>
> | **Metric**               | **CogVideoX-2B** | **Wan-1.3B** | **TTM + Wan-1.3B** |
> | ------------------------ | ---------------- | ------------ | ------------------ |
> | imaging_quality ↑        | 0.4870           | 0.6851       | **0.6877**         |
> | motion_smoothness ↑      | 0.9886           | 0.9874       | **0.9935**         |
> | dynamic_degree ↑         | 0.3465           | 0.3960       | **0.4059**         |
> | subject_consistency ↑    | 0.9347           | 0.9324       | **0.9529**         |
> | background_consistency ↑ | 0.9585           | 0.9375       | **0.9628**         |
> | aesthetic_quality ↑      | 0.3849           | **0.5370**   | 0.4924             |
> | hf_clip_vit_b32 ↑        | **0.2428**       | 0.2234       | 0.2288             |
> | hf_clip_vit_l14 ↑        | **0.2007**       | 0.1788       | 0.1892             |
> | hf_laion_clip_vit_b32 ↑  | 0.1859           | 0.1791       | **0.1980**         |
>
>
> We now elaborate on why we adopt the GAN framework as the primary implementation in our work. The key difference between our GAN-based framework and diffusion-based approaches lies mainly in the training objective rather than the architecture. To illustrate, consider an ideal scenario where both videos and their true latent factors are observable. In this fully supervised case, the model could be trained in a straightforward, sample-to-sample manner. However, under the unsupervised setting considered in this work, the latent variables are unobserved, and the model must instead ensure observational equivalence, i.e., aligning the distributions of generated and real videos. This naturally motivates the use of a GAN framework, which enforces such distributional alignment.
>
> Importantly, the theoretical guarantees established in our framework are also compatible with diffusion-based generative processes. In practice, one potential solution is to adopt a GAN-style loss for matching the marginal video distributions while retaining a diffusion-based architecture. Alternatively, it is possible to predict the independent noise component directly from video sequences, enabling sample-to-sample training that preserves extra supervision signals, as demonstrated in [1]. The latter one is used in our experiments in Appendix F.3.
>
> We have added this discussion about GAN and diffusion models to Appendix G (line 1388) in the revised paper for completeness.
>
> [1] Learning Vision and Language Concepts for Controllable Image Generation.

---

> ### Author Response · Authors · 2025-11-23
> **Part II**
>
> >**W2:** Experiments only focus on domain-specific datasets, how will be the performance of CoVoGAN on UCF101 or even larger dataset such as Kinetics? It would be better if the author could show the abilities of the model on more complex datasets.
>
> **A2:** We thank you for your valuable suggestion. We have now included experiments on the UCF101 dataset in Appendix F.8 (line 1268).
>
> Techinically, we evaluate our method on 3 subsets, with 3 actions, 20 actions and all 101 actions, to evaluate the performance under different levels of complexity. Since it is impractical to use keypoints as ground truth for controllability evaluation (like what we did in the main experiments) when multiple actions are involved, we instead assess controllability by measuring the classification accuracy of generated videos with respect to their target action classes. We calculate the FVD to assess the visual fidelity of the generated videos. Here we attach the results for your reference. The baseline models crashed when training on 20 and more classes, so we only report MoStGAN-V results for 3 classes. The performance drop when increasing the number of classes is reasonable, as many of the actions in UCF101 are not object-centric, which is challenging for GAN based models. The performance drop when increasing the number of classes is reasonable, as UCF-101 does not contain enough video samples per class, as discussed in [1].
>
> | Class No. | Metric     | FVD ↓   | Accuracy ↑ |
> |:----------:|:-----------|:--------|:------------|
> | **3**      | MoStGAN-V  | 897.86  | 0.35 |
> | **3**      | CoVoGAN    | 178.03  | 0.64 |
> | **20**     | CoVoGAN    | 790.23  | - |
> | **101**    | CoVoGAN    | 1050.61 | - |
>
> [1] Adversarial Video Generation on Complex Datasets.
>
> >**W3:** Can CoVoGAN be used in long-term video generation?
>
> **A3:** Yes. Since our TTM module is designed in an autoregressive manner, CoVoGAN can be naturally extended to long-term video generation. We would like to respectfully note that we have already conducted video generation experiments with different sequence lengths, including 16 and 32 frames, as we have reported in Appendix F.1 in the original submission (line 1130 in the revised version). Here we attach the results for your reference. It worth noting that as the sequence length increases, the generation task becomes more challenging, leading to a general decline in performance across all models. Nevertheless, CoVoGAN consistently outperforms baseline in both short-term and long-term video generation scenarios.
>
> | **Metrics with 16 frames**      | **MoStGAN-V** | **CoVoGAN** |
> | ---------------- | ------------- | ----------- |
> | FVD ↓            | 49.85        | **48.8**  |
> | MCC (%) ↑        | 29.00         | **33.78**   |
> | SAP (%) ↑        | 4.25          | **8.48**    |
> | Modularity (%) ↑ | 7.66          | **17.37**   |
>
> | **Metrics with 32 frames**      | **MoStGAN-V** | **CoVoGAN** |
> | ---------------- | ------------- | ----------- |
> | FVD ↓            | 159.24        | **145.77**  |
> | MCC (%) ↑        | 14.23         | **29.99**   |
> | SAP (%) ↑        | 0.87          | **4.20**    |
> | Modularity (%) ↑ | 8.15          | **10.14**   |

---

### Official Review · Reviewer_9LG2 · 2025-11-09

**Soundness:** 3
**Presentation:** 3
**Contribution:** 2
**Rating:** 6
**Confidence:** 2

**Summary:**

This paper presents Controllable Video Generative Adversarial Networks , a novel framework designed to address the challenge of fine-grained control in video generation by leveraging provable disentanglement. The authors introduce a Temporal Transition Module that separates static content elements and dynamic style dynamics, grounded in theoretical guarantees of block-wise and component-wise identifiability. By integrating TTM into a GAN architecture, CoVoGAN achieves disentangled representations through the "minimal change principle" and "sufficient change property," enabling independent manipulation of motion components like head movement and eye blinking. Extensive experiments on datasets including FaceForensics and RealEstate demonstrate superior performance in both generation quality  and controllability  compared to baselines.

**Strengths:**

The paper provides a strong theoretical foundation by proving block-wise and component-wise identifiability under mild assumptions. This formalizes why disentanglement between motion and content is achievable, addressing gaps in prior heuristic-based methods. The TTM effectively combines GRU for temporal dependency modeling and Deep Sigmoid Flow for conditional independence, ensuring dynamic variables are disentangled and controllable. Ablation studies validate the critical role of GRU and flow components in maintaining performance.

**Weaknesses:**

While CoVoGAN excels in unsupervised disentanglement, it does not address how to align latent dimensions with semantic labels (e.g., explicitly mapping a dimension to "eye blinking"). Integrating weak supervision or text guidance could enhance interpretability and usability.
Additionally, the article lacks experiments on the UCF101 dataset, and it is recommended to supplement them.

**Questions:**

The method in this paper belongs to unconditional video generation. Can it be applied to text-to-video generation models? The latter has greater practical application value.

---

> ### Author Response · Authors · 2025-11-23
>
> Dear Reviewer 9LG2, thank you for the constructive comments, which helped us enhance the semantic alignment discussion and improve the completeness of our experiments.
>
> >**W1 and Q1:** While CoVoGAN excels in unsupervised disentanglement, it does not address how to align latent dimensions with semantic labels (e.g., explicitly mapping a dimension to "eye blinking"). Integrating weak supervision or text guidance could enhance interpretability and usability ... The method in this paper belongs to unconditional video generation. Can it be applied to text-to-video generation models? The latter has greater practical application value.
>
> **A1:** Thank you for the insightful suggestion. We fully agree that integrating weak supervision or textual guidance can further enhance the interpretability and semantic alignment of the latent dimensions. In fact, our formulation naturally extends to such a setting. When a supervised or conditional signal $u$ (e.g., text prompt or trajectory) is available, the generative process in Equation 1 can be written as:
>
> $$
> z _{t,i}^s = f _i^s\big(\mathbf{Pa}(z _{t,i}^s), \epsilon _{t,i}^s, u \big), \quad
> z _j^c = f _j^c\big(\epsilon _j^c, u \big),
> $$
>
> where $u$ provides an additional supervision signal. As stated in Theorem 2 of [1], the identifiability results are easier to establish, because the changes of $u$ can help to better satisfy Assumption C2.
>
> We have revised first paragraph of Section 3 (line 181) and added Appendix A.4 (line 1004) in the revised manuscript to discuss this extension.
>
> To demonstrate the flexibility of this extension, we have incorporated experiments in Appendix F.3 in the original submission (line 1172 in the revised paper) using both text-guided (TTM + Wan-1.3B) and trajectory-guided (TTM + MotionCtrl) settings. These results verify that the proposed Temporal Transition Module (TTM) can serve as a general, theoretically grounded plugin for conditional video generation, improving controllability and interpretability.
>
> We have also added UCF-101 as a conditional generation experiment in Appendix F.9 (line 1289) which incorporates action labels as a weak supervision signal. Please see Answer A2 for more details.
>
> [1] Temporally Disentangled Representation Learning.
>
> >**W2:** Additionally, the article lacks experiments on the UCF101 dataset, and it is recommended to supplement them.
>
> **A2:** We appreciate your valuable suggestion. We have now included experiments on the UCF101 dataset in Appendix F.8 (line 1268).
>
> Techinically, we evaluate our method on 3 subsets, with 3 actions, 20 actions and all 101 actions, to evaluate the performance under different levels of complexity. Since it is impractical to use keypoints as ground truth for controllability evaluation (like what we did in the main experiments) when multiple actions are involved, we instead assess controllability by measuring the classification accuracy of generated videos with respect to their target action classes. We calculate the FVD to assess the visual fidelity of the generated videos. Here we attach the results for your reference. The baseline models crashed when training on 20 and more classes, so we only report MoStGAN-V results for 3 classes. The performance drop when increasing the number of classes is reasonable, as UCF-101 does not contain enough video samples per class, as discussed in Appendix C.1 in [2].
>
> | Class No. | Metric     | FVD ↓   | Accuracy ↑ |
> |:----------:|:-----------|:--------|:------------|
> | **3**      | MoStGAN-V  | 897.86  | 0.35 |
> | **3**      | CoVoGAN    | 178.03  | 0.64 |
> | **20**     | CoVoGAN    | 790.23  | - |
> | **101**    | CoVoGAN    | 1050.61 | - |
>
>
> [2] Adversarial Video Generation on Complex Datasets.

---

### Author Response · Authors · 2025-12-03
**Concise Summary of Rebuttal Responses**

Dear Area Chair, Reviewers, and Readers,

We sincerely thank the area chair and all reviewers for their thoughtful feedback and the time and effort dedicated to evaluating our work. We also appreciate the reviewers who recognized the strengths of our paper, including its strong (9LG2, KQV9), solid (7TFq), and rigorous (S4AE) theoretical foundation, its novel (9LG2) framework, and its efficiency (S4AE). Below, we summarize the main concerns raised and our corresponding responses.

Methodological and Theoretical Clarifications

1. As suggested by Reviewer 9LG2 (W1 and Q1), we show that our theory naturally extends to conditional generation. A detailed discussion has been added to Appendix A.4.
2. We provide a detailed explanation of why we choose GANs as our backbone model (Appendix G). We also demonstrate that our framework can be incorporated into a diffusion model, with supporting evidence in Appendix F.3 (Reviewer S4AE W1, Reviewer 7TFq W3).
3. To address Reviewer KQV9’s concern (W1) that our content–motion independence assumption may be too strong, we provide further analysis in Appendix A.5: (i) we show that independence typically holds, including in the examples cited by the reviewer; (ii) even in extreme cases where the two factors are entangled, such scenarios can still be interpreted within our framework via appropriate transformations.
4. We clarify why we compare disentanglement metrics against diffusion models, despite their lack of a compact latent space (Reviewer 7TFq W1).

Additional Experiments

1. Following the suggestion of Reviewers 9LG2 (W2) and S4AE (W2), we add experiments on UCF-101 in Appendix F.8 to improve completeness.
2. We clarify that our model supports long-term video generation, with empirical evidence provided in Appendix F.1 (Reviewer S4AE W3).
3. To address Reviewer KQV9’s concern (W2) regarding performance in highly dynamic scenes, we analyze performance gains across datasets and show that our model is in fact better suited for highly dynamic scenarios. We also report consistent gains on higher-resolution datasets.
4. Based on Reviewer 7TFq’s suggestion (W2), we add an Average Keypoint Distance (AKD) comparison in Appendix F.9 to explicitly evaluate blockwise disentanglement.

We again thank all reviewers for their insightful comments, which have greatly helped us improve the clarity, completeness, and overall quality of our paper. We are grateful for the time and effort each reviewer dedicated to strengthening our work.



Sincerely,

The Authors of Submission 12470

---

### Meta-Review · Area_Chair_vEeF · 2026-01-06

**Summary:**

- This paper proposes a controllable video generator with a temporal module for provable disentanglement via block-wise (content vs motion) and component-wise (within motion) identifiability.
- Reviewers largely agreed the paper is well-written / easy to follow, with solid theory and extensive experiments showing FVD and controllability, plus ablations supporting GRU/flow.
- The rebuttal added diffusion/conditional results, UCF101, and new quantification of content–motion independence, the paper is suitable for poster acceptance.

**Reviewer Concerns:**

Concerns addressed:
- Unconditional -> conditional  text-to-video: authors state the formulation naturally extends with supervised/conditional signals, add TTM+Wan-1.3B and TTM+MotionCtrl, and discuss GAN vs diffusion
- Missing UCF101: authors add UCF-101 experiments, reporting FVD and accuracy as controllability proxy
- Quantify content–motion independence: authors add AKD keypoint distance when altering content latent, showing stronger disentanglement.
- Long-term video generation: authors point to autoregressive TTM and include 16 vs 32 frame results (already in appendix)

Outstanding concerns:
- Semantic alignment: remains mainly extension rather than a full solution for explicitly labeling latent dimensions.
- Generality / broader domains: concerns about “limited generalizability” and performance on “more complex datasets” are mitigated but not fully settled

**Reviewer Scores:**

- 9LG2 likely to stay at 6
- S4AE likely moves 4 -> 6 (diffusion integration, UCF101, long-term addressed, but still cautious on fidelity/complex datasets)
- KQV9 likely to move  4 -> 6 (assumption discussion, added UCF-style completeness helps, but generalizability concern remains)
- 7TFq likely stays at 6

---

### Decision · Program_Chairs · 2026-01-26

Accept (Poster)